# ERAlign: Energy-based Representation Alignment of GNNs and LLMs on Text-attributed Graphs

Xianlin Zeng [1]   Fan Xia [2]   Xiangyu Chen [1]

## Abstract

Text-attributed Graphs (TAGs) incorporate textual node attributes with graph structures to describe rich relational semantics. Recent efforts to integrate Graph Neural Networks (GNNs) and Large Language Models (LLMs) have shown promise for learning on TAGs, yet achieving well-aligned representations remains challenging. Prior studies largely rely on heuristics that perform coarse-grained matching. They lack sufficient constraints and ignore distributional alignment, leading to representation drift and limited generalization. Building on Energy-based Models (EBMs), we propose an **E**nergy-based **R**epresentation **Align**ment (ERAlign) framework that projects GNN-encoded graph structure and LLM-derived text embeddings in a shared latent space to achieve distribution consistency. Concretely, layer-wise alignment is quantified by a distance metric and optimized via an EBM objective. By decreasing energy values, our framework yields well-aligned representations for downstream tasks. During training, we introduce Energy Discrepancy (ED) to avoid high sampling costs associated with intractable normalization. ED also carries theoretical guarantees of higher training efficiency and reduced energy landscape distortion. Empirical evaluations on eight TAG datasets demonstrate that ERAlign obtains state-of-the-art performance across varying levels of supervision and cross-task transfer scenarios.

[1]Postdoctoral Research Station at China RongTong Academy of Sciences Group Corporation Limited, Beijing, China [2]Duke University, Durham, NC, USA. Correspondence to: Xianlin Zeng <divinezeng@gmail.com>.

*Proceedings of the $43^{rd}$ International Conference on Machine Learning*, Seoul, South Korea. PMLR 306, 2026. Copyright 2026 by the author(s).

## 1. Introduction

Text-attributed Graphs (TAGs) (Yang et al., 2021; Jin et al., 2023), in which each node is associated with a piece of text (e.g., a document, profile, or description), are ubiquitous in domains such as academic citation networks (Mernyei & Cangea, 2020; Sinha et al., 2015), social networks (Kim et al., 2020; Huang et al., 2024; Newman et al., 2002), and e-commerce graphs (McAuley et al., 2015; Hu et al., 2020). Effective learning on TAGs necessitates combining node-level textual semantics with structural information from edges. Graph Neural Networks (GNNs) (Kipf & Welling, 2017; Hamilton et al., 2017; Veličković et al., 2018) excel at modeling graph topology via message passing, while Large Language Models (LLMs) (Devlin et al., 2019; Radford et al., 2018; Touvron et al., 2023) capture rich linguistic knowledge and support reasoning. Therefore, synergizing these complementary paradigms has emerged as a natural progression. Indeed, recent studies (Li et al., 2024; Wang et al., 2025) have shown that enhancing graph learning with LLMs yields significant performance improvements. However, accurately aligning the representations of GNNs and LLMs faces three critical limitations: (i) Insufficient constraints. Training relies primarily on proxy signals such as autoregressive token loss or synthetic pseudo pairs of embeddings and texts (Zhang et al., 2024; Huang et al., 2024). (ii) Coarse-grained matching. Alignment is typically restricted to the output stage via logit mixing or concatenation, failing to handle intermediate representations (Zhao et al., 2023; Jin et al., 2023). (iii) Inconsistent distribution. Existing methods emphasize local sample similarity but overlook global distributional consistency, rendering the latent space vulnerable to distribution shifts (Wang et al., 2024).

**Motivation**. The aforementioned methods often suffer from suboptimal alignment between token semantics and structural cues, hindering cross-modal synergy and degrading downstream performance. We draw inspiration from Energy-based Models (EBMs) (LeCun et al., 2006; Grathwohl et al., 2020; Xie et al., 2016; Zhao et al., 2017), a probabilistic framework for representation learning successfully applied to image synthesis (Du & Mordatch, 2019; Guo et al., 2023), text generation (Bakhtin et al., 2021; Xu et al., 2025), and more recently graph learning (Chen et al.,

2020; Zeng et al., 2025; Chen et al., 2025). EBMs define an unnormalized density via a learnable parametric network that assigns lower energy to higher probability mass, thereby enhancing distributional consistency. We leverage this property as a principled constraint to align representations, enabling efficient bidirectional interaction between GNNs and LLMs. Additionally, EBMs support flexible training strategies grounded in solid theory (Hinton, 2002; Song & Kingma, 2021).

**Contribution**. In this paper, we propose a novel **E**nergy-based **R**epresentation **Align**ment (ERAlign) framework on TAGs that projects GNN-encoded structural signals and LLM-derived textual semantics into a shared latent space and enforces distributional consistency. Specifically, we unify the dimensionality of GNN and LLM representations and match their intermediate layers to achieve layer-wise alignment. We enable bidirectional information fusion by injecting intermediate LLM semantics into GNN message passing and feeding graph representations into LLMs via soft prompts. Rather than modeling the data distribution directly, we formulate a set EBM over the latent representations. The EBM objective quantifies representation misalignment using the Cramér distance (Bellemare et al., 2017), which captures the geometric structure of the distributions to mitigate cross-modality drift, complementing the inherent task-specific supervision. To avoid the expensive sampling required by standard EBM training, we introduce an Energy Discrepancy (ED) minimization scheme. ED contrasts the energy potentials of data samples with noise-perturbed samples at an appropriate scale. We further propose a multi-scale estimation strategy to bridge distant global modes while preserving fine-grained local structure in the energy landscape, along with a stabilization term to improve training robustness. To support diverse tasks, our framework incorporates two alternative output heads, yielding corresponding variants including $\text{ERAlign}_{\text{GNN}}$ and $\text{ERAlign}_{\text{LLM}}$. Extensive experiments on eight benchmark datasets demonstrate that ERAlign outperforms state-of-the-art methods. Beyond standard evaluations, ERAlign exhibits strong label efficiency in semi-supervised settings and promising zero-shot cross-task transferability.

The major contributions are threefold as follows:

- We propose a novel ERAlign framework that aligns GNN-encoded structural representations with LLM-derived textual representations within a shared latent space. Layer-wise alignment is quantified via the Cramér distance and optimized using an EBM objective, aiming to enhance distributional consistency.
- We introduce an ED training scheme that requires no additional sampling iterations beyond standard EBM optimization. ED also enjoys theoretical guarantees for improved training efficiency and stronger global-local modeling via

multi-scale estimation strategy.
- Extensive experiments on eight benchmark datasets show that ERAlign substantially outperforms state-of-the-art methods. We further develop two variants to support evaluations in different supervision ratios and zero-shot cross-task transfer scenarios.

## 2. Related Work

### 2.1. Graph Energy-based Model

Recent studies have extended EBMs to diverse graph-related downstream tasks. GraphEBM (Chen et al., 2020) constructs an edge probability space and uses an EBM to learn a latent sampling distribution over candidate edges, producing task-specific and interpretable affinity graphs. ECL-GSR (Zeng et al., 2025) combines EBMs with contrastive learning to approximate the joint distribution of augmented graph views, pruning the adjacency matrix according to learned similarity and improving node classification. GAD-EBM (Roy et al., 2023) learns the energy over ego-subgraphs via a subgraph score matching objective in a discrete state space, and takes the induced likelihoods as anomaly scores. DeGEM (Chen et al., 2025) decouples a topology-aware graph encoder from an energy head in latent space and trains EBMs with approximate maximum likelihood, enhancing node out-of-distribution detection on heterophilic graphs. Collectively, these methods leverage EBMs to define scalar energies or likelihood scores for graph generation, structure learning, and anomaly detection.

### 2.2. GNN-LLM Integration

Existing techniques fall into two categories: LLM-as-enhancer and LLM-as-predictor. Enhancers typically treat LLMs as semantic feature extractors and inject the resulting embeddings into graph pipelines. For instance, TAPE (He et al., 2024) utilizes LLM reasoning to generate explanatory text, which an interpreter module translates into compact features for subsequent GNN. ENGINE (Zhu et al., 2024) preserves a frozen LLM backbone while injecting structural signals through lightweight side structures, employing a caching mechanism to accelerate inference. OFA (Liu et al., 2024) addresses cross-domain generalization by unifying heterogeneous data into TAGs and establishing a graph prompting paradigm for in-context learning. In contrast, predictors couple graph structures directly with LLM decoding for inference. LLaGA (Chen et al., 2024) reorganizes nodes into structure-aware sequences projected into the token embedding space, thereby allowing instruction tuning without backbone modification. Similarly, GraphAdapter (Huang et al., 2024) leverages a GNN as an adapter to infuse topology-aware context into a frozen LLM. It proposes an autoregressive next-token prediction objective to facilitate downstream adaptation through task prompts. TEA-

GLM (Wang et al., 2024) aligns GNN representations with LLM embeddings by learning a linear projector that converts graph data into soft tokens. These tokens are integrated into unified instructions for cross-task zero-shot learning. Unlike previous works, we project GNN and LLM representations into a shared latent space for alignment, offering two variants to cover both enhancer and predictor paradigms.

## 3. Preliminaries

**Text-attributed Graphs (TAGs)** are graphs whose nodes are associated with textual descriptions in addition to the graph structure. Formally, a TAG is defined by $\mathcal{G} = (\mathcal{V}, \mathcal{E}, \mathbf{A}, \mathcal{S}_\mathcal{V})$, where $\mathcal{V} = \{v_1, \ldots, v_N\}$ is the set of $N$ nodes, $\mathcal{E} \subseteq \mathcal{V} \times \mathcal{V}$ is the edge set, and $\mathbf{A} \in \{0,1\}^{|\mathcal{V}| \times |\mathcal{V}|}$ is the adjacency matrix with $\mathbf{A}_{ij} = 1$ iff $(v_i, v_j) \in \mathcal{E}$. Each node $v$ has a corresponding textual attribute $s_v$, and we denote the collection of all node texts by $\mathcal{S}_\mathcal{V} = \{s_v\}_{v \in \mathcal{V}}$.

**Energy-based Models (EBMs)** define the probability density over continuous variables $\boldsymbol{x}$ via a learnable scalar energy function $E_\theta(\boldsymbol{x})$. The density is represented as the Boltzmann distribution:

$$p_\theta(\boldsymbol{x}) = \frac{\exp(-E_\theta(\boldsymbol{x}))}{Z_\theta}, \ Z_\theta = \int \exp(-E_\theta(\boldsymbol{x}))\mathrm{d}\boldsymbol{x}, \quad (1)$$

where $Z_\theta$ is the partition function ensuring normalization. Since computing $Z_\theta$ is generally intractable in high dimensions, existing methods operate directly on unnormalized densities, relying mainly on Contrastive Divergence (CD) or Score Matching (SM) (Song & Kingma, 2021).

Standard EBM training minimizes CD between the data distribution $p_\mathrm{d}$ and the model distribution $p_\theta$, equivalently maximizing the log-likelihood $\mathcal{L}(\theta)$. The gradient of $\mathcal{L}(\theta)$ decomposes into positive and negative phases:

$$\nabla_\theta \mathcal{L}(\theta) = \mathbb{E}_{\boldsymbol{x} \sim p_\mathrm{d}}[\nabla_\theta E_\theta(\boldsymbol{x})] - \mathbb{E}_{\tilde{\boldsymbol{x}} \sim p_\theta}[\nabla_\theta E_\theta(\tilde{\boldsymbol{x}})], \quad (2)$$

where the first term lowers the energy of real data, while the second term raises the energy of generated samples. Evaluating the second expectation requires sampling $\tilde{\boldsymbol{x}}$ from $p_\theta$, typically approximated via Markov Chain Monte Carlo (MCMC) techniques such as Langevin Dynamics.

SM proposes to match the model score $\nabla_{\boldsymbol{x}} E_\theta(\boldsymbol{x})$ with data score $\nabla_{\boldsymbol{x}} \log p_\mathrm{d}(\boldsymbol{x})$ by minimizing the Fisher Divergence:

$$\mathcal{F}_{\mathrm{SM}}(\theta) = \mathbb{E}_{\boldsymbol{x} \sim p_\mathrm{d}}\left[\frac{1}{2}\|-\nabla_{\boldsymbol{x}} E_\theta(x) - \nabla_{\boldsymbol{x}} \log p_\mathrm{d}(\boldsymbol{x})\|^2\right]. \quad (3)$$

## 4. Method

In this section, we detail the proposed ERAlign framework. We first describe the overall architecture including layer-wise alignment and output heads. Then, we utilize the

Cramér distance to quantify misalignment and develop a set EBM for optimization. Finally, we present the ED minimization scheme and joint training objective.

### 4.1. Architecture Overview

As illustrated in Fig. 1, ERAlign achieves layer-wise alignment using a dual-stream encoder with projection heads. We employ a $\mathcal{K}$-layer GNN $g_\theta$ that captures structural semantic relations. Given a TAG $\mathcal{G}$, $g_\theta$ processes the adjacency matrix $\mathbf{A}$ and node set $\mathcal{V}$ to produce embeddings $\mathbf{H}_k^\mathcal{G} \in \mathbb{R}^{|\mathcal{V}| \times d_\mathcal{G}}$ at layer $k$. In parallel, a $\mathcal{J}$-layer pretrained LLM $f_\theta$ encodes the textual attributes $s_v$ of each node $v$ into embeddings $\mathbf{H}_j^\mathcal{T} \in \mathbb{R}^{|\mathcal{V}| \times d_\mathcal{T}}$ at layer $j$.

**Layer-wise Alignment.** Since model depth $\mathcal{K} \ll \mathcal{J}$ and dimensionality $d_\mathcal{G} \ll d_\mathcal{T}$, direct alignment is infeasible. We address the depth difference by pairing each GNN layer with LLM layers selected at fixed intervals. Let $(k, j) \in \mathcal{P}$ denote the set of aligned layer indices. This design preserves cross-modal constraints while maintaining a manageable memory and computational cost. To tackle dimension mismatch, we utilize learnable projectors $\pi_\mathcal{G}$ and $\pi_\mathcal{T}$ to map representations into a unified dimension $\mathbb{R}^{d_\mathcal{A}}$. Specifically, $\pi_\mathcal{G}$ projects the low-dimensional GNN embeddings upward, while $\pi_\mathcal{T}$ projects the high-dimensional LLM embeddings downward. We define the paired representations as:

$$\mathbf{Z}_k^\mathcal{G} = \pi_\mathcal{G}(\mathbf{H}_k^\mathcal{G}), \quad \mathbf{Z}_j^\mathcal{T} = \pi_\mathcal{T}(\mathbf{H}_j^\mathcal{T}). \quad (4)$$

We inject intermediate LLM semantics into GNN before the next message passing step. At each aligned layer, a projector $\tilde{\pi}_\mathcal{G}$ maps LLM representations into GNN embeddings:

$$\tilde{\mathbf{H}}_k^\mathcal{G} = (1 - \alpha)\mathbf{H}_k^\mathcal{G} + \alpha\tilde{\pi}_\mathcal{G}(\mathbf{Z}_j^\mathcal{T}), \quad (5)$$

where $\alpha$ balances the fusion strength. As LLM layers progressively abstract textual attributes into high-level semantics, GNN layers expand their structural receptive fields through successive message passing. This layer-wise injection grounds structural aggregation in the LLM's semantic space, facilitating signal propagation across the topology.

**Output Head.** To support diverse downstream tasks, ERAlign adopts two alternative output heads comprising a linear classifier and a prompt-based interface, which connect to GNN and LLM backbones, respectively.

For supervised node classification, ERAlign$_{\mathrm{GNN}}$ projects the final-layer GNN embedding $\mathbf{H}_\mathcal{K}^\mathcal{G}$ to class probabilities over the label set using a softmax function.

To exploit the semantic reasoning capabilities of LLMs, ERAlign$_{\mathrm{LLM}}$ incorporates structural information by injecting the aligned GNN embeddings as soft prompts into the instruction. Similarly, we employ a projector $\tilde{\pi}_\mathcal{T}$ to map

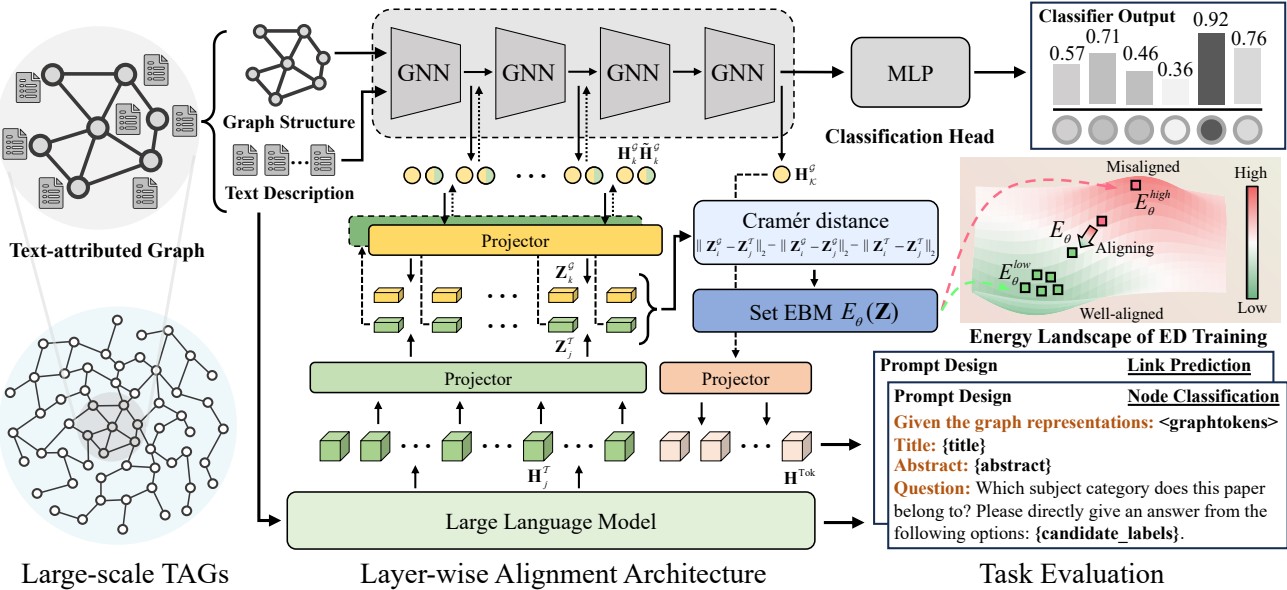

*Figure 1.* Overview of ERAlign for energy-based representation alignment between GNNs and LLMs on text-attributed graphs.

the final-layer embedding $\mathbf{H}_{\mathcal{K}}^{\mathcal{G}}$ back to $B$ token embeddings $\{\mathbf{H}^{\text{Tok}}\}_B$. For node classification, we construct prompts with a candidate label set and compute each label score by the conditional log-likelihood of its verbalizer. For link prediction, we formulate a binary query for node pairs, deriving edge scores from the log-probability difference between affirmative and negative verbalizers. In this way, ERAlign$_{\text{LLM}}$ improves generalization under label scarcity and facilitates cross-task transfer to unseen graphs.

## 4.2. Energy-based Representation Alignment

A standard measure of similarity between probability distributions is the Kullback-Leibler (KL) divergence. However, it relies solely on density ratios and ignores the geometric scale of the outcome space. Here, geometry refers not to the graph topology but to the continuous metric space in which the embeddings reside, where representation distances directly encode semantic similarity. The Wasserstein distance captures such geometry, yet its sample gradients are biased in high dimensions, rendering naive stochastic gradient descent unreliable. We therefore adopt the Cramér distance, which combines robust geometric sensitivity with low-variance sample gradients.

Beyond the choice of metric, the alignment objective itself is critical. Objectives such as the InfoNCE loss enforce only point-wise similarity between paired samples. Since GNNs and LLMs encode heterogeneous signals, purely local matching is prone to a failure mode in which individual pairs align while the underlying distributions remain misaligned. We therefore embed the Cramér distance within an EBM to encourage global distribution alignment.

**Definition 1** Given a set of latent representations $\mathbf{Z} = \{\mathbf{z}_i^{\mathcal{G}}, \mathbf{z}_i^{\mathcal{T}}\}_{i=1}^N$, we formulate the unnormalized density via a set EBM as $p_\theta(\mathbf{Z}) \propto \exp(-E_\theta(\mathbf{Z}))$.

The set energy $E_\theta(\mathbf{Z})$ parameterized by $\theta$ is a scalar defined by the empirical Cramér distance:

$$E_\theta(\mathbf{Z}) = 2\,\widehat{\mathbb{E}}_{i,j}\big[\|\mathbf{z}_i^{\mathcal{G}} - \mathbf{z}_j^{\mathcal{T}}\|_2\big] \\ - \widehat{\mathbb{E}}_{i,j}\big[\|\mathbf{z}_i^{\mathcal{G}} - \mathbf{z}_j^{\mathcal{G}}\|_2 + \|\mathbf{z}_i^{\mathcal{T}} - \mathbf{z}_j^{\mathcal{T}}\|_2\big], \tag{6}$$

where $\widehat{\mathbb{E}}_{i,j}$ denotes the empirical expectation $\frac{1}{N^2}\sum_{i,j}$. The first term minimizes cross-modal distance to align topologies with textual semantics, whereas the latter two terms maximize intra-modal dispersion to prevent trivial collapse.

Minimizing $E_\theta$ penalizes misalignment by assigning lower energy to latent sets $\mathbf{Z}$ where cross-modal distances are small relative to intra-modal dispersions. Consequently, the EBM acts as a layer-wise regularizer that mitigates modality drift and improves robustness under distribution shift.

## 4.3. Energy Discrepancy Minimization

Training EBMs with CD typically requires a short-run Langevin sampler. However, it is computationally expensive and sensitive to mixing across modes, often destabilizing training. SM avoids sampling but suffers from nearsightedness as its objective relies solely on local gradients, failing to capture global proportions. To overcome these limitations, we introduce an ED training scheme.

**Theoretical Analysis.** Considering a random perturbation process $q(\cdot|\mathbf{Z})$, we perturb $\mathbf{Z}$ with an isotropic noise at scale $t$ to obtain perturbed sample $\tilde{\mathbf{Z}}_t$.

**Definition 2** Let $q(\tilde{\mathbf{Z}}_t|\mathbf{Z})$ be a conditional probability density, contrastive potential $E_q$ induced by $q$ is defined as:

$$E_q(\tilde{\mathbf{Z}}_t) = -\log \int q(\tilde{\mathbf{Z}}_t|\mathbf{Z}) \exp(-E_\theta(\mathbf{Z})) \mathrm{d}\mathbf{Z}. \quad (7)$$

ED is defined as the discrepancy between $E_\theta$ and $E_q$:

$$\mathrm{ED}_q = \mathbb{E}_{p_d(\mathbf{Z})}[E_\theta(\mathbf{Z})] - \mathbb{E}_{p_d(\mathbf{Z})}\mathbb{E}_{q(\tilde{\mathbf{Z}}_t|\mathbf{Z})}[E_q(\tilde{\mathbf{Z}}_t)]. \quad (8)$$

Intuitively, Eq. 8 contrasts the energy of the true data manifold with that of perturbed samples at an appropriate noise scale. More importantly, this objective constitutes a valid gradient field for updating $\theta$, thereby eliminating the need for MCMC sampling and reducing the time complexity.

**Theorem 1** Under Gaussian perturbation, the ED objective induces a gradient field that interpolates between score matching updates as $t$ approaches 0 and likelihood gradients as $t$ becomes larger.

It demonstrates that ED bridges score matching and maximum likelihood estimation. With $t > 0$, the perturbation smooths the data distribution, enabling ED to capture global structure rather than merely local variations. However, a single noise scale implies that large $t$ bridges distant modes but blurs fine structure, whereas small $t$ preserves detail but cannot fully overcome the nearsightedness. Accordingly, we define a multi-scale form of ED objective by integrating over $t \in (0, T]$.

**Theorem 2** Let $\gamma(\tilde{\mathbf{Z}}_t|\mathbf{Z})$ be a Gaussian transition density, the ED objective can be written in multi-scale form:

$$\mathcal{L}_{\mathrm{ED}_\gamma} = \int_0^T \mathbb{E}_{p_d(\mathbf{Z})} \mathbb{E}_{\gamma(\tilde{\mathbf{Z}}_t|\mathbf{Z})} \left[ \frac{1}{2} \|\nabla_{\tilde{\mathbf{Z}}_t} E_\gamma(\tilde{\mathbf{Z}}_t)\|^2 - \Delta_{\tilde{\mathbf{Z}}_t} E_\gamma(\tilde{\mathbf{Z}}_t) \right] \mathrm{d}t. \quad (9)$$

**Implementation.** In Theorem 2, the expectation within the contrastive potential $E_\gamma$ is analytically intractable. We therefore employ a Monte Carlo estimator based on the Gaussian conditional distribution $\gamma = \mathcal{N}(\tilde{\mathbf{Z}}_t|\mathbf{Z}, t\mathbf{I})$. Exploiting the symmetric kernel form $\gamma(\tilde{\mathbf{Z}}_t|\mathbf{Z}) = \gamma(\mathbf{Z}|\tilde{\mathbf{Z}}_t)$, we approximate $E_\gamma$ by following the forward perturbation. At the $i$-th scale $t_i$, we generate $M$ perturbed samples $\{\tilde{\mathbf{Z}}_{t_i,j}\}_{j=1}^M$ via $\tilde{\mathbf{Z}}_{t_i,j} = \tilde{\mathbf{Z}}_{t_i} + \sqrt{t_i}\boldsymbol{\xi}_j$, where $\boldsymbol{\xi}_j \sim \mathcal{N}(\mathbf{0}, \mathbf{I})$. The contrastive potential is estimated as:

$$E_\gamma(\tilde{\mathbf{Z}}_{t_i}) \approx -\log\left(\frac{1}{M} \sum_{j=1}^M \exp(-E_\theta(\tilde{\mathbf{Z}}_{t_i,j}))\right). \quad (10)$$

However, substituting Eq. 10 directly into Eq. 9 can cause numerical instability and optimization difficulties when $M$ is small, as the logarithmic estimator is biased and yields high-variance gradients. To mitigate this, we introduce a

---

**Algorithm 1** ERAlign Training Algorithm

**Input:** TAG $\mathcal{G}$, GNN $g_\theta$ with $\mathcal{K}$ layers, LLM $f_\theta$ with $\mathcal{J}$ layers, projectors $\pi_\mathcal{G}$, $\pi_\mathcal{T}$, $\tilde{\pi}_\mathcal{G}$, $\tilde{\pi}_\mathcal{T}$, $\mathcal{P}$ aligned indices, $S$ scales.
**Output:** Trained ERAlign framework (updated parameters $\theta$).
**Initialization:** Initialize parameters $\theta$ with $\alpha$, $w$, $\lambda$, $M$, and $d_\mathcal{A}$.
**while** *not converged* **do**
    Compute embeddings $\{\mathbf{H}_k^\mathcal{G}\}_\mathcal{K}$ and $\{\mathbf{H}_j^\mathcal{T}\}_\mathcal{J}$ with $g_\theta$ and $f_\theta$;
    **for** $(k, j) \in \mathcal{P}$ **do**
        Compute representations $\mathbf{Z}_k^\mathcal{G}$ and $\mathbf{Z}_j^\mathcal{T}$ with Eq. 4;
        Compute embedding $\tilde{\mathbf{H}}_k^\mathcal{G}$ with $\mathbf{Z}_j^\mathcal{T}$ and $\mathbf{H}_k^\mathcal{G}$ using Eq. 5;
        Calculate $E_\theta(\mathbf{Z}_k^\mathcal{G}, \mathbf{Z}_j^\mathcal{T})$ with Cramér distance via Eq. 6;
        **for** $t_i \in S$ **do**
            Generate $M$ perturbations $\tilde{\mathbf{Z}}_{t_i}$ with Gaussian noise;
            Estimate contrastive potential $E_\gamma(\tilde{\mathbf{Z}}_{t_i})$ via Eq. 10;
            Accumulate $w$-stabilized objective $\mathcal{L}_{\mathrm{ED}}$ with Eq. 11;
    Project embedding $\mathbf{H}_\mathcal{K}^\mathcal{G}$ to $B$ embeddings $\mathbf{H}^{\mathrm{Tok}}$ using $\tilde{\pi}_\mathcal{T}$;
    Minimize joint objective $\mathcal{L}_{\mathrm{total}}$ in Eq. 12 by gradient descent;

---

$w$-stabilization term, $w/M \exp(-E_\theta(\mathbf{Z}))$. This term imposes a deterministic upper bound on the logarithm argument, preventing divergence when perturbed samples fall into high-energy regions. Thus, this approach reduces both the estimator's variance and the sample size required for stable training. Formally, given a sequence of corresponding perturbed samples $\{\tilde{\mathbf{Z}}_{t_i,j}\}_{j=1}^M$ spanning $S$ scales, the ED objective is approximated by:

$$\mathcal{L}_{\mathrm{ED}}(\theta) \approx \frac{1}{S} \sum_{i=1}^S \log\left(\frac{w}{M} + \frac{1}{M} \sum_{j=1}^M \exp(E_\theta(\mathbf{Z}) - E_\theta(\tilde{\mathbf{Z}}_{t_i,j}))\right). \quad (11)$$

A suitable weight $w > 0$ provides numerical stability.

**Training Objective.** During training, the task-specific loss $\mathcal{L}_{\mathrm{task}}$ is tailored to the variant, including standard cross-entropy for ERAlign$_{\mathrm{GNN}}$ and verbalizer-based cross-entropy for ERAlign$_{\mathrm{LLM}}$. Simultaneously, ED losses are minimized to update the GNN and LLM backbones along with all projections. The overall training objective is:

$$\mathcal{L}_{\mathrm{total}} = \mathcal{L}_{\mathrm{task}} + \lambda \sum_{(k,j) \in \mathcal{P}} \mathcal{L}_{\mathrm{ED}}^{(k,j)}(\theta), \quad (12)$$

where $\mathcal{P}$ is the aligned indices and $\lambda$ is the hyperparameter. Algorithm 1 outlines the detailed training procedure.

## 5. Experiments

We conduct extensive experiments to evaluate ERAlign in terms of effectiveness, robustness, and transferability. Specifically, we seek to answer the following research questions: **RQ1**: How does ERAlign perform on node classification in fully-supervised and semi-supervised settings? **RQ2**: How well does ERAlign transfer to link prediction in the zero-shot scenario? **RQ3**: How do the components

*Table 1.* Node classification performance under the full-supervised setting with accuracy±std (%) reported. **Bold** and underline highlight the best and second-best results, respectively.

| CATEGORY | METHODS | CORA | CITESEER | PUBMED | ARXIV | INSTAGRAM | REDDIT | PHOTO | COMPUTER |
|---|---|---|---|---|---|---|---|---|---|
| **GNNs** | MLP | 71.57 ± 1.56 | 72.36 ± 1.30 | 87.26 ± 0.24 | 58.02 ± 0.30 | 62.48 ± 1.11 | 61.23 ± 0.34 | 61.21 ± 0.11 | 82.60 ± 0.60 |
| | GCN | 86.36 ± 0.93 | 76.44 ± 0.79 | 89.12 ± 0.38 | 73.91 ± 0.21 | 67.60 ± 0.82 | 64.73 ± 0.61 | 78.34 ± 0.39 | 77.16 ± 3.80 |
| | GAT | 87.26 ± 1.51 | 76.96 ± 1.12 | 89.26 ± 0.44 | 73.82 ± 0.14 | 67.13 ± 0.89 | 64.39 ± 0.57 | 81.55 ± 0.35 | 88.32 ± 0.24 |
| | GraphSAGE | 87.10 ± 1.43 | 77.07 ± 1.09 | 89.30 ± 0.28 | 73.39 ± 0.25 | 68.16 ± 0.71 | 64.82 ± 0.55 | 79.14 ± 0.26 | 87.77 ± 0.34 |
| **PLMs** | BERT | 79.61 ± 1.40 | 74.06 ± 1.26 | 90.95 ± 0.11 | 72.66 ± 0.24 | 60.11 ± 0.93 | 58.70 ± 0.54 | 70.01 ± 0.08 | 74.22 ± 0.21 |
| | DeBERTa | 77.79 ± 2.26 | 73.13 ± 1.94 | 90.81 ± 0.20 | 73.61 ± 0.12 | 62.40 ± 0.59 | 59.92 ± 0.45 | 70.18 ± 0.18 | 74.82 ± 0.16 |
| | RoBERTa | 80.35 ± 0.48 | 75.90 ± 1.69 | 91.13 ± 0.11 | 74.12 ± 0.12 | 63.75 ± 1.13 | 60.61 ± 0.24 | 79.45 ± 0.37 | 75.76 ± 0.30 |
| **Enhancers** | TAPE | 88.05 ± 1.76 | 76.45 ± 1.60 | 90.38 ± 0.99 | 74.96 ± 0.14 | 65.44 ± 0.35 | 63.01 ± 0.82 | 82.26 ± 0.64 | − |
| | GLEM | 87.07 ± 1.01 | 76.30 ± 2.45 | 89.56 ± 1.65 | 73.55 ± 0.22 | 65.90 ± 0.36 | 60.88 ± 0.03 | 77.74 ± 0.27 | 81.63 ± 0.46 |
| | GIANT | 87.58 ± 0.46 | 77.25 ± 0.77 | 88.58 ± 0.22 | 72.62 ± 0.11 | 59.86 ± 0.22 | 63.79 ± 0.45 | − | − |
| | SimTeG | 88.75 ± 0.42 | 77.37 ± 0.64 | 88.31 ± 0.75 | 77.48 ± 0.16 | 64.29 ± 0.19 | 61.60 ± 0.88 | 79.82 ± 0.21 | 85.25 ± 0.06 |
| | ENGINE | 86.79 ± 0.58 | 75.82 ± 1.52 | 90.08 ± 0.16 | 74.69 ± 0.36 | 66.27 ± 0.41 | 62.57 ± 0.13 | 83.06 ± 0.22 | − |
| **Predictors** | LLaGA | 85.54 ± 0.96 | 72.95 ± 1.48 | 86.58 ± 0.94 | 74.49 ± 0.23 | 67.25 ± 0.87 | 64.56 ± 0.48 | 87.62 ± 0.30 | 87.75 ± 0.12 |
| | GraphGPT | 85.12 ± 1.25 | 73.32 ± 1.60 | 87.64 ± 0.73 | 75.15 ± 0.14 | 65.79 ± 1.22 | 60.72 ± 1.47 | 84.46 ± 0.36 | 86.78 ± 1.14 |
| | GraphAdapter | 88.95 ± 0.97 | 77.52 ± 1.58 | 91.39 ± 0.42 | 77.07 ± 0.15 | 65.13 ± 0.75 | 64.61 ± 0.19 | 83.18 ± 0.31 | − |
| **Ours** | ERAlign$_{\text{LLM}}$ | 89.20 ± 1.12 | 77.82 ± 0.50 | 89.55 ± 1.32 | 76.81 ± 0.78 | 68.05 ± 0.73 | 64.95 ± 0.51 | 86.95 ± 0.28 | 88.12 ± 0.66 |
| | ERAlign$_{\text{GNN}}$ | **90.75 ± 0.76** | **78.54 ± 0.34** | **92.17 ± 0.29** | **78.07 ± 0.15** | **68.66 ± 0.55** | **65.58 ± 0.28** | **89.38 ± 0.21** | **90.07 ± 0.18** |

*Table 2.* Node classification performance under the semi-supervised setting with accuracy±std (%) reported. **Bold** and underline highlight the best and second-best results, respectively.

| METHODS | CORA | CITESEER | INSTAGRAM | PHOTO |
|---|---|---|---|---|
| MLP | 30.41 ± 0.59 | 27.74 ± 0.59 | 62.54 ± 1.02 | 46.21 ± 0.61 |
| GCN | 41.96 ± 0.73 | 30.53 ± 0.68 | 63.20 ± 0.31 | 51.27 ± 0.33 |
| GAT | 38.86 ± 0.59 | 30.50 ± 0.27 | 63.17 ± 0.86 | 50.35 ± 0.85 |
| GraphSAGE | 37.60 ± 0.67 | 31.69 ± 0.44 | 61.70 ± 0.93 | 52.17 ± 0.65 |
| BERT | 37.59 ± 0.08 | 31.50 ± 0.54 | 62.50 ± 0.43 | 49.59 ± 0.04 |
| DeBERTa | 29.98 ± 1.09 | 30.80 ± 0.55 | 63.59 ± 0.27 | 47.96 ± 1.47 |
| RoBERTa | 28.23 ± 0.00 | 23.32 ± 4.55 | 63.77 ± 0.52 | 49.52 ± 0.12 |
| TAPE | 47.08 ± 0.20 | 29.77 ± 0.28 | 61.25 ± 0.59 | 59.76 ± 0.12 |
| GLEM | 49.01 ± 0.58 | 36.64 ± 1.46 | 61.54 ± 0.56 | 56.25 ± 2.14 |
| SimTeG | 45.78 ± 0.22 | 30.40 ± 0.66 | 60.61 ± 0.16 | 55.73 ± 0.84 |
| ENGINE | 42.32 ± 0.66 | 35.70 ± 0.19 | 63.88 ± 0.20 | 57.96 ± 0.13 |
| ERAlign$_{\text{LLM}}$ | 51.97 ± 0.83 | 37.85 ± 1.05 | 63.96 ± 0.95 | 61.82 ± 0.91 |
| ERAlign$_{\text{GNN}}$ | **52.33 ± 0.71** | **38.12 ± 0.63** | **64.80 ± 0.22** | **63.27 ± 0.35** |

and hyperparameters contribute to ERAlign? **RQ4**: How efficient is ERAlign during training? **RQ5**: What kind of aligned representations does ERAlign learn?

## 5.1. Experimental Setups

**Datasets.** We evaluate ERAlign on eight TAG benchmarks across citation networks (Cora, CiteSeer, PubMed (Sen et al., 2008), and Arxiv (Hu et al., 2020)), social media (Huang et al., 2024) (Reddit and Instagram), and e-commerce platforms (Yan et al., 2023) (Computer and Photo). Within these graphs, nodes represent papers, users, and products, while edges encode citation links, social interactions, and co-purchasing relationships, respectively. Comprehensive dataset statistics are provided in Appendix A.2.

**Baselines.** We compare against four groups: (1) GNNs (MLP, GCN, GAT, GraphSAGE), (2) Pre-trained Language Models (PLMs) (BERT (Devlin et al., 2019), RoBERTa (Liu et al., 2019), DeBERTa (He et al., 2021)), (3) GNN–LLM Enhancers (OFA (Liu et al., 2024), GIANT (Chien et al., 2022), TAPE (He et al., 2024), GLEM (Zhao et al., 2023), SimTeG (Duan et al., 2023), ENGINE (Zhu et al., 2024)), (4) GNN–LLM Predictors (LLaGA (Chen et al., 2024), GraphGPT (Tang et al., 2024), GraphAdapter (Huang et al., 2024), TEA-GLM (Wang et al., 2024)). All evaluations are performed across 10 different random seeds. We report the mean accuracy with standard deviation for node classification and mean AUC for link prediction.

**Implementation Details.** All experiments are conducted using PyTorch and PyTorch Geometric on two NVIDIA RTX 5090 GPUs. For a fair comparison, we adopt the standard train/validation/test splits under the fully supervised setting, as detailed in Table 8 of Appendix A.2. For the semi-supervised setting, we sample only 20% of the labeled training nodes while preserving the full graph structure and the original validation and test sets. Following (Zhang et al., 2026), this setting is intentionally stricter than the standard Planetoid split, thereby inducing an extreme few-shot and class-imbalanced scenario. ERAlign integrates a $\mathcal{K}$=4 layer GraphSAGE with hidden dimension $d_{\mathcal{G}}$=256 and a $\mathcal{J}$=32 layer LLaMA2-7B fine-tuned using LoRA (Hu et al., 2022). All projectors are 2-layer MLPs with GELU activation. The projection dimension $d_{\mathcal{A}}$ is 512. Aligned layer indices $\mathcal{P}$ are $\{(1, 4), (2, 12), (3, 20), (4, 28)\}$. We employ $S$=4 noise scales over $[0.01, 10.0]$ with $M$=4 perturbed samples. Hyperparameters are set to $\alpha$=0.5, $w$=0.1, and $\lambda$=1.0. AdamW is configured with a learning rate of $10^{-3}$, weight decay

of $5 \times 10^{-5}$, and batch size of 256. We utilize a cosine annealing scheduler with a 5-epoch warm-up and train for a maximum of 200 epochs with an early stopping.

## 5.2. Node Classification Performance (RQ1)

**Full-supervised Setting.** As presented in Table 1, ERAlign$_{GNN}$ achieves the state-of-the-art across all eight benchmark datasets, significantly outperforming GNN baselines, text-only PLMs, and prior GNN–LLM integration methods. On citation networks, our method exceeds the strongest baselines, GraphAdapter and SimTeG, by margins of 0.6–1.8%. In the social media domain, ERAlign$_{GNN}$ improves over LLaGA by approximately 1.0–1.4%. Notably, on e-commerce platforms, the performance gap is even more pronounced, with substantial gains of 1.8% over the second-best methods. The consistent gains across diverse domains indicate that well-aligned representations within a shared latent space and energy-based objective effectively match graph structure with textual semantics during feature propagation, leading to higher performance. ERAlign$_{LLM}$ also delivers highly competitive results, particularly on social media domain. A comparison of the two variants suggests that ERAlign$_{GNN}$ remains as the primary discriminator in label-rich scenarios.

**Semi-supervised Setting.** As reported in Table 2, ERAlign achieves superior performance in the semi-supervised setting. Specifically, ERAlign$_{GNN}$ consistently outperforms baselines across four datasets. Relative to the leading integrations, ERAlign$_{GNN}$ obtains substantial margins by 3.3% on Cora, 1.4% on CiteSeer, and 3.5% on Photo, highlighting excellent label efficiency. The results indicate that ERAlign effectively injects semantic knowledge from LLMs to structure learning in GNNs, thereby mitigating dependency on extensive annotations. Moreover, the narrowing performance gap between ERAlign variants indicates that our framework can better leverage the reasoning capabilities of LLMs. As a result, ERAlign is validated as an effective graph learner under label-constrained scenarios.

## 5.3. Cross-task Transferability (RQ2)

To evaluate the cross-task transferability of ERAlign in the zero-shot scenario, we directly apply the model trained only for node classification to link prediction, without any additional fine-tuning. As shown in Table 3, ERAlign$_{LLM}$ achieves the highest AUC scores across all five datasets, substantially outperforming the strongest zero-shot transfer baseline, TEA-GLM, by a margin of 1.2–3.2%. This suggests that ERAlign promotes robust representation alignment, which is sufficiently task-agnostic to mitigate distribution shifts while remaining discriminative for node semantics, facilitating effective transfer from node classification to edge inference on unseen graphs.

*Table 3.* Link prediction performance under the zero-shot scenario with AUC (%) reported. Models are transferred from node classification. **Bold** highlights the best result.

| METHODS | CORA | PUBMED | ARXIV | PHOTO | COMPUTER |
|---|---|---|---|---|---|
| OFA | 49.20 | 48.10 | 46.90 | 45.90 | 46.10 |
| LLaGA | 53.70 | 56.90 | 57.00 | 47.80 | 47.90 |
| GraphGPT | 52.00 | 50.10 | 64.90 | – | – |
| TEA-GLM | 58.60 | 68.90 | 65.70 | 54.50 | 55.40 |
| ERAlign$_{LLM}$ | **60.45** | **71.17** | **66.91** | **57.13** | **58.63** |

## 5.4. Ablation Study (RQ3)

We perform a series of ablation experiments to validate the design effectiveness of ERAlign.

*Table 4.* Node classification accuracy (%) of ERAlign$_{GNN}$ on Cora, PubMed, and Photo datasets using different alignment intervals. **Bold** indicates the best performance.

| STRATEGY | LAYER INDEX | CORA | PUBMED | PHOTO |
|---|---|---|---|---|
| Output only | $\{32\}$ | 87.14 | 85.22 | 86.36 |
| Sparse interval | $\{4, 16, 28\}$ | 88.96 | 87.88 | 88.41 |
| Medium interval | $\{4, 12, 20, 28\}$ | **90.75** | 92.17 | **89.38** |
| Dense interval | $\{4, 8, \ldots, 28, 32\}$ | 90.21 | **92.47** | 89.03 |

**Alignment Strategy.** We align each GNN layer with selected LLM layers at different intervals and compare four strategies. As shown in Table 4, the medium interval delivers the best accuracy on Cora and Photo, while the dense interval achieves the highest performance on PubMed. Output-only alignment fails to correct intermediate representation drift, whereas sparse interval provides insufficient cross-modal constraints. Conversely, dense interval may introduce excessive computational overhead or compromise the LLM's semantic richness, thereby reducing generalization. Consequently, the medium interval offers the optimal trade-off between accuracy and efficiency.

*Table 5.* Node classification accuracy (%) of ERAlign$_{GNN}$ on Cora, PubMed, and Photo datasets with different metrics and objectives. **Bold** indicates the best performance.

| METRIC | OBJECTIVE | CORA | PUBMED | PHOTO |
|---|---|---|---|---|
| Cosine distance | CL via InfoNCE | 87.52 | 89.19 | 87.14 |
| Wasserstein distance | OT via Sinkhorn | 88.10 | 90.45 | 88.40 |
| Euclidean distance | EBM via CD | 86.95 | 88.20 | 86.55 |
|  | EBM via ED | 87.32 | 89.46 | 87.92 |
| Cramér distance | EBM via CD | 90.50 | 91.25 | 89.12 |
|  | EBM via SM | 88.35 | 91.05 | 88.85 |
|  | EBM via ED | **90.75** | **92.17** | **89.38** |

**Alternative Metrics and Objectives.** We evaluate the impact of different distance metrics and training objectives. For cosine similarity, we use Contrastive Learning (CL) with an InfoNCE objective. We adopt Optimal Transport

(OT) with Sinkhorn iterations for Wasserstein distance. Table 5 demonstrates that Cramér distance variants outperform the alternatives on Cora, PubMed, and Photo, underscoring the advantage of distribution statistics over Euclidean distance. Cosine distance underperforms on larger graphs, suggesting that purely relational similarity is insufficient. Wasserstein distance is theoretically appealing but computationally expensive. Under the same metric, CD and SM achieve slightly lower classification performance compared to ED.

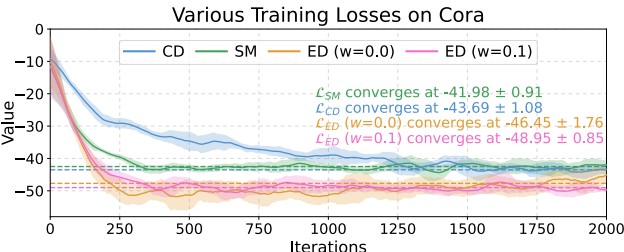

*Figure 2.* Training dynamics of ERAlign against multiple losses and $w$-stabilization settings on Cora dataset.

**Training Dynamics.** We analyze the convergence behavior of various training objectives in Fig. 2 by plotting loss curves over 2000 iterations on Cora. We observe that CD exhibits slow convergence and high variance, stemming from the additional Langevin sampling steps. While SM converges rapidly, it is sensitive to the noise scale and stabilizes at a notably higher final value. The naive ED with $w$=0.0 suffers from estimator bias and numerical instability, resulting in severe oscillations and high variance. Finally, Fig. 2 illustrates that the proposed stabilized ED with $w$=0.1 achieves superior convergence with reduced variance, validating the effectiveness of the $w$-stabilization term.

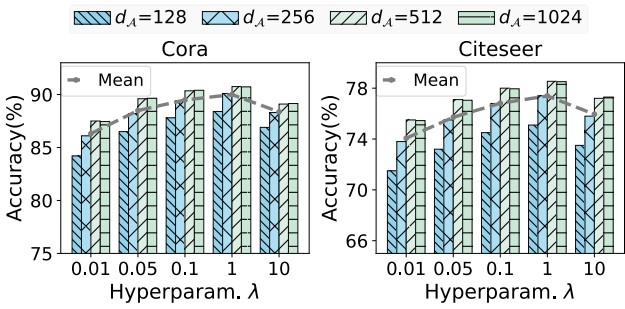

*Figure 3.* Hyperparameter $\lambda$ and dimensionality $d_{\mathcal{A}}$ analysis of ERAlign on Cora and CiteSeer datasets.

**Hyperparameter Sensitivity.** We study the sensitivity of ERAlign to key hyperparameters on Cora and CiteSeer. Fig. 3 reports accuracy under varying weight $\lambda$ and dimension $d_{\mathcal{A}}$. Performance typically peaks at $\lambda$=1.0. Smaller values lead to under-alignment, while overly large values

over-constrain task-relevant discrimination. Regarding $d_{\mathcal{A}}$, a size of 512 provides sufficient representation capacity. Too small dimensions cause information bottlenecks that hurt accuracy, whereas larger dimensions increase computational complexity with limited gains. Fig. 4 displays the impact of noise scale $t$ sets $\{\{0.01\}, \{10.0\}, \{0.01, 10.0\}, \{0.1, 1.0\}, \{0.01, 0.1, 1.0, 10.0\}\}$ and perturbed samples $M \in \{1, 2, 4, 8, 16\}$. According to Theorem 1, $t$ controls the nearsightedness of ED. Analysis shows that small $t$ prevents ED from resolving local mixture weights, leading to degraded performance. Larger $t$ mitigates this issue but increases the variance of the loss estimator, which theoretically necessitates a larger $M$. Empirically, ERAlign with $M$=4 attains near-peak performance, and gains plateau beyond 8, making larger values inefficient.

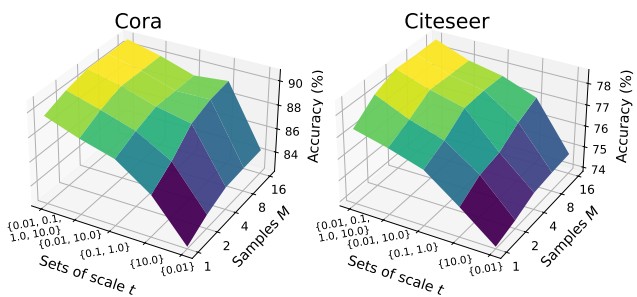

*Figure 4.* Noise scale $t$ sets and perturbed samples $M$ analysis of ERAlign on Cora and CiteSeer datasets.

We further ablate the fusion strength $\alpha$, which balances the GNN representation against the injected LLM semantics. As reported in Table 6, the optimal $\alpha$ is only mildly dataset-dependent. ERAlign consistently attains peak or near-peak accuracy for intermediate values, with $\alpha$=0.5 yielding the best on Cora and CiteSeer and near-best on PubMed. Too small an $\alpha$ under-utilizes the injected LLM semantics, whereas too large an $\alpha$ over-suppresses the GNN's structural signal, with both damaging performance.

*Table 6.* Node classification accuracy (%) of ERAlign$_{\text{GNN}}$ under varying fusion strength $\alpha$. **Bold** indicates the best performance.

| STRENGTH $\alpha$ | CORA | CITESEER | PUBMED |
|---|---|---|---|
| 0.05 | 87.52 | 77.10 | 89.95 |
| 0.25 | 90.11 | 78.32 | 91.50 |
| 0.50 | **90.75** | **78.54** | 92.17 |
| 0.75 | 90.10 | 78.25 | **92.32** |
| 0.95 | 89.27 | 77.78 | 91.73 |

### 5.5. Training Efficiency (RQ4)

Table 7 summarizes training efficiency on the Arxiv dataset. Leveraging LoRA and lightweight projectors, ERAlign updates only 11.3M parameters by an approximate 10× reduction compared to full fine-tuning baselines like GIANT and GLEM. ERAlign completes training in roughly 9 hours

with 16.3GB peak memory, achieving speedups of $3.3\times$ and $5.2\times$ over GIANT and GLEM, respectively, while also outperforming LLaGA. Although ENGINE exhibits lower latency due to the frozen LLM backbone, it suffers from sub-optimal performance. ERAlign provides a more favorable balance between task accuracy and training efficiency.

*Table 7.* The training efficiency of different methods on the Arxiv dataset. Total training time is reported using two RTX 5090 GPUs.

| METHODS | PARAM. (M) | MEMORY (GB) | TOTAL TIME |
|---|---|---|---|
| GLEM | 138.6 | 18.1 | 48h 14m |
| GIANT | 114.5 | 20.6 | 30h 31m |
| LLaGA | 19.6 | 17.4 | 10h 51m |
| ENGINE | 3.9 | 14.9 | 5h 17m |
| ERAlign | 11.3 | 16.3 | 9h 11m |

We also analyze the computational complexity of ERAlign. Although the empirical Cramér distance requires pairwise distance computation, it is evaluated within each mini-batch rather than across the full node set. Given a batch size $B$ and projection dimension $d_{\mathcal{A}}$, the per-step cost of pairwise alignment is bounded by $\mathcal{O}(B^2 \cdot d_{\mathcal{A}})$. This cost reduces to highly parallelizable matrix multiplications on GPUs and thus incurs only minimal overhead. Moreover, by eliminating the inner Langevin sampling loops required by CD, the ED scheme lowers the optimization complexity by roughly an order of magnitude. Consequently, the dominant cost of ERAlign in practice stems from the LoRA fine-tuning of the LLaMA2-7B backbone.

### 5.6. Visualization (RQ5)

We employ t-SNE to project the embeddings $H^{\mathcal{G}}$ and $H^{\mathcal{T}}$ in Cora dataset into a 2D plane, visualizing their distributions in the shared latent space. Points are colored by layer indices and distinguished by modality in Fig. 5. The resulting well-separated clusters exhibit high intra-class compactness, confirming that ERAlign preserves semantic discriminability across layers. Furthermore, the visualization highlights a substantially reduced modality gap, as GNN and LLM embeddings are effectively aligned within mixed clusters rather than separated into disjoint distributions.

## 6. Conclusion

We develop an ERAlign framework that aligns layer-wise representations of GNNs and LLMs with an EBM objective, enforcing distribution consistency. We further introduce an ED scheme to avoid high sampling costs while ensuring global-local modeling and efficient training. Empirical evaluations highlight ERAlign's superior performance in node classification under different supervision settings and its robust zero-shot transferability for link prediction.

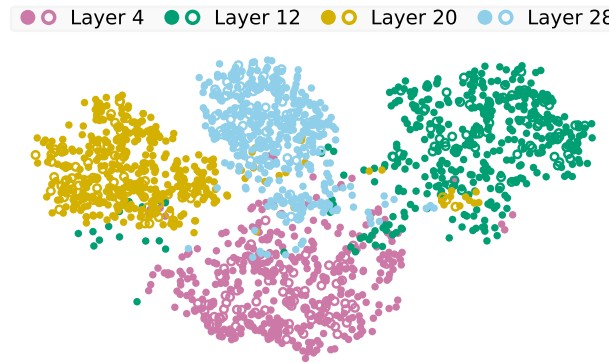

*Figure 5.* 2D t-SNE visualizations on Cora dataset. Solid and hollow points denote GNN and LLM embeddings, respectively.

## Acknowledgements

We thank the anonymous reviewers for their valuable feedback and constructive suggestions. This work was supported by the China Postdoctoral Science Foundation under Grant Number 2025M771684.

## Impact Statement

This paper aims to advance the field of graph machine learning by proposing a GNN–LLM integration framework for learning on TAGs. Deploying ERAlign in real-world applications, such as social media analysis and e-commerce recommendation, requires careful ethical consideration. In these domains, textual node attributes may contain sensitive personally identifiable information, so practical deployments should enforce rigorous data anonymization and strict access control. Moreover, since ERAlign aligns graph structures with LLM-derived text embeddings, it may inherit or amplify social biases present in the LLM pretraining corpora. Practitioners should therefore audit model outputs for fairness prior to deployment.

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

# A. Appendix

## A.1. Proofs and derivations

In this section, we present the proofs and derivations for the Energy Discrepancy (ED) within our framework. We first formally define the ED and clarify its theoretical properties. Then, we derive its connections to score matching and maximum likelihood. Finally, we detail the practical implementation, including Monte Carlo estimation and the $w$-stabilization term.

**Definition of Contrastive Potential and Energy Discrepancy.** Let $p_d(\mathbf{Z})$ denote the data distribution over latent representations. We define an EBM with the unnormalized density $p_\theta(\mathbf{Z}) \propto \exp(-E_\theta(\mathbf{Z}))$, where $E_\theta(\mathbf{Z})$ is a learnable energy function. Given a perturbation kernel $q(\tilde{\mathbf{Z}}_t|\mathbf{Z})$ (e.g., Gaussian noise) that generates a perturbed sample $\tilde{\mathbf{Z}}_t$ from $\mathbf{Z}$ at scale $t$, we define the contrastive potential $E_q(\tilde{\mathbf{Z}}_t)$ as the negative log-density of the distribution smoothed by $q$:

$$E_q(\tilde{\mathbf{Z}}_t) = -\log \int q(\tilde{\mathbf{Z}}_t|\mathbf{Z})\exp(-E_\theta(\mathbf{Z}))\mathrm{d}\mathbf{Z}. \tag{13}$$

Essentially, $E_q$ signifies the energy landscape smoothed via convolution with the kernel $q$. Importantly, $E_q$ is not an independently parameterized function but is derived entirely from $E_\theta$. ED is defined as the difference between the expected energy of real data and the expected contrastive potential of perturbed data.

ED is defined as the discrepancy between $E_\theta$ and $E_q$:

$$\mathrm{ED}_q = \mathbb{E}_{p_\mathrm{d}(\mathbf{Z})}[E_\theta(\mathbf{Z})] - \mathbb{E}_{p_\mathrm{d}(\mathbf{Z})}\mathbb{E}_{q(\tilde{\mathbf{Z}}_t|\mathbf{Z})}[E_q(\tilde{\mathbf{Z}}_t)]. \tag{14}$$

Unlike standard EBM training, ED utilizes a known perturbation kernel $q$ to formulate $E_q$, whose gradients can be estimated directly from forward sampling. As $E_q$ relies solely on $E_\theta$, it avoids both the normalization constant or score function gradients. Consequently, optimizing $\mathrm{ED}_q$ only requires sampling from $p_d$ and $q$, eliminating the need for expensive MCMC sampling during training and enhancing computational efficiency.

**Connection to Maximum Likelihood Estimation.** We establish the theoretical foundation for Theorem 1 by explicating the relationship between the ED and Maximum Likelihood Estimation (MLE). We elucidate the relationship between ED and Score Matching in the context of Theorem 2.

**Theorem 1** Under Gaussian perturbation $\gamma_t(\tilde{\mathbf{Z}}|\mathbf{Z}) = \mathcal{N}(\tilde{\mathbf{Z}}|\mathbf{Z}, t\mathbf{I})$, the gradient field of the ED objective interpolates between score matching updates as $t \to 0$ and maximum likelihood gradients as $t \to \infty$.

Let the normalized EBM density be defined as $p_\theta(\mathbf{Z}) = \exp(-E_\theta(\mathbf{Z}))/Z_\theta$, where $Z_\theta = \int \exp(-E_\theta(\mathbf{Z}))\mathrm{d}\mathbf{Z}$. Consider the smoothed data distribution $p_{t,\mathrm{d}}(\tilde{\mathbf{Z}}_t) = \int p_\mathrm{d}(\mathbf{Z})\gamma(\tilde{\mathbf{Z}}_t|\mathbf{Z})\mathrm{d}\mathbf{Z}$ and the unnormalized smoothed model density defined by the contrastive potential:

$$\tilde{p}_{t,\theta}(\tilde{\mathbf{Z}}_t) = \exp(-E_\gamma(\tilde{\mathbf{Z}}_t)) = \int \exp(-E_\theta(\mathbf{Z}))\gamma(\tilde{\mathbf{Z}}_t|\mathbf{Z})\mathrm{d}\mathbf{Z}. \tag{15}$$

The corresponding normalized smoothed model distribution is $p_{t,\theta}(\tilde{\mathbf{Z}}_t) = \tilde{p}_{t,\theta}(\tilde{\mathbf{Z}}_t)/Z_\theta$. Note that the partition function $Z_\theta$ remains invariant under the convolution with the normalized kernel $\gamma$. Consequently, the log-density of the smoothed model is given by:

$$\log p_{t,\theta}(\tilde{\mathbf{Z}}_t) = -E_\gamma(\tilde{\mathbf{Z}}_t) - \log Z_\theta. \tag{16}$$

Recall the definition of the ED objective under the perturbation kernel $\gamma$:

$$\mathcal{L}_{\mathrm{ED}_{\gamma(t)}}(\theta) = \mathbb{E}_{p_\mathrm{d}(\mathbf{Z})}[E_\theta(\mathbf{Z})] - \mathbb{E}_{p_{t,\mathrm{d}}(\tilde{\mathbf{Z}}_t)}[E_\gamma(\tilde{\mathbf{Z}}_t)]. \tag{17}$$

Simultaneously, the expected log-likelihood of the original data is:

$$\mathcal{L}_{\mathrm{MLE}}(\theta) = \mathbb{E}_{p_\mathrm{d}(\mathbf{Z})}[\log p_\theta(\mathbf{Z})] = \mathbb{E}_{p_\mathrm{d}(\mathbf{Z})}[-E_\theta(\mathbf{Z})] - \log Z_\theta. \tag{18}$$

We analyze the KL divergence between the smoothed data and smoothed model distributions:

$$\mathrm{KL}(p_{t,\mathrm{d}}\|p_{t,\theta}) = \mathbb{E}_{p_{t,\mathrm{d}}}[\log p_{t,\mathrm{d}}(\tilde{\mathbf{Z}}_t) - \log p_{t,\theta}(\tilde{\mathbf{Z}}_t)]$$
$$= \underbrace{\mathbb{E}_{p_{t,\mathrm{d}}}[\log p_{t,\mathrm{d}}(\tilde{\mathbf{Z}}_t)]}_{-\mathcal{H}(p_{t,\mathrm{d}})} - \mathbb{E}_{p_{t,\mathrm{d}}}[\log p_{t,\theta}(\tilde{\mathbf{Z}}_t)]. \tag{19}$$

Let $\mathcal{H}(p_{t,\mathrm{d}}) = \mathbb{E}_{p_{t,\mathrm{d}}}[\log p_{t,\mathrm{d}}(\tilde{\mathbf{Z}}_t)]$ be the entropy of the smoothed data distribution. Substituting Eq. 16 into the second term yields:

$$\mathrm{KL}(p_{t,\mathrm{d}} \| p_{t,\theta}) = -\mathcal{H}(p_{t,\mathrm{d}}) + \mathbb{E}_{p_{t,\mathrm{d}}}[E_\gamma(\tilde{\mathbf{Z}}_t)] + \log Z_\theta. \tag{20}$$

From Eq. 17 and Eq. 18, we can substitute $\mathbb{E}_{p_{t,\mathrm{d}}}[E_\gamma(\tilde{\mathbf{Z}}_t)]$ and $\log Z_\theta$:

$$\begin{aligned} \mathrm{KL}(p_{t,\mathrm{d}} \| p_{t,\theta}) &= -\mathcal{H}(p_{t,\mathrm{d}}) + \big(\mathbb{E}_{p_\mathrm{d}}[E_\theta(\mathbf{Z})] - \mathcal{L}_{\mathrm{ED}}(\theta)\big) + \big(-\mathbb{E}_{p_\mathrm{d}}[E_\theta(\mathbf{Z})] - \mathcal{L}_{\mathrm{MLE}}(\theta)\big) \\ &= -\mathcal{H}(p_{t,\mathrm{d}}) - \mathcal{L}_{\mathrm{ED}_\gamma(t)}(\theta) - \mathcal{L}_{\mathrm{MLE}}(\theta). \end{aligned} \tag{21}$$

Rearranging terms, we obtain the identity:

$$\mathcal{L}_{\mathrm{ED}_\gamma(t)}(\theta) = -\mathcal{L}_{\mathrm{MLE}}(\theta) - \mathrm{KL}(p_{t,\mathrm{d}} \| p_{t,\theta}) - \mathcal{H}(p_{t,\mathrm{d}}). \tag{22}$$

By the following information-transport inequality (Raginsky & Sason, 2013), the KL divergence between the smoothed distributions is bounded by the Wasserstein-2 distance $\mathcal{W}_2$ between the original distributions:

$$\mathrm{KL}(p_{t,\mathrm{d}} \| p_{t,\theta}) \leq \frac{1}{2t} \mathcal{W}_2^2(p_\mathrm{d}, p_\theta). \tag{23}$$

Assuming $\mathcal{W}_2(p_\mathrm{d}, p_\theta)$ is finite, the KL term vanishes as $t \to \infty$. Taking the gradient of Eq. 22 w.r.t. $\theta$, where $\mathcal{H}(p_{t,\mathrm{d}})$ is independent of $\theta$, we have:

$$\nabla_\theta \mathcal{L}_{\mathrm{ED}_\gamma(t)}(\theta) = -\nabla_\theta \mathcal{L}_{\mathrm{MLE}}(\theta) - \nabla_\theta \mathrm{KL}(p_{t,\mathrm{d}} \| p_{t,\theta}). \tag{24}$$

As $t \to \infty$, $\nabla_\theta \mathrm{KL}(p_{t,\mathrm{d}} \| p_{t,\theta}) \to 0$. Consequently, minimizing the ED becomes asymptotically equivalent to maximizing the log-likelihood:

$$\lim_{t \to \infty} \nabla_\theta \mathcal{L}_{\mathrm{ED}_\gamma(t)}(\theta) = -\nabla_\theta \mathcal{L}_{\mathrm{MLE}}(\theta). \tag{25}$$

**Relationship with Score Matching.** We provide the detailed proof of Theorem 2.

**Theorem 2** Let $\gamma(\tilde{\mathbf{Z}}_t|\mathbf{Z})$ be the Gaussian transition density. The ED objective is equivalent to the integral of the score matching objective over scale $t$:

$$\mathcal{L}_{\mathrm{ED}_\gamma(t)} = \int_0^T \mathbb{E}_{p_\mathrm{d}(\mathbf{Z})} \mathbb{E}_{\gamma(\tilde{\mathbf{Z}}_t|\mathbf{Z})} \Big[\frac{1}{2} \|\nabla_{\tilde{\mathbf{Z}}_t} E_\gamma(\tilde{\mathbf{Z}}_t)\|^2 - \Delta_{\tilde{\mathbf{Z}}_t} E_\gamma(\tilde{\mathbf{Z}}_t)\Big] \mathrm{d}t. \tag{26}$$

The Gaussian transition density $\gamma_t(\tilde{\mathbf{Z}}|\mathbf{Z}) = \mathcal{N}(\tilde{\mathbf{Z}}|\mathbf{Z}, t\mathbf{I})$ satisfies the heat equation $\partial_t \gamma = \frac{1}{2}\Delta_{\tilde{\mathbf{Z}}_t}\gamma$. We analyze the derivative of the ED w.r.t. the noise scale $t$. Recall the definition of the contrastive potential in Eq. 15, we observe that $\exp(-E_\gamma)$ is the convolution of $\exp(-E_\theta)$ with the Gaussian kernel $\gamma$. Since convolution commutes with differential operators, $\exp(-E_\gamma)$ also satisfies the heat equation:

$$\partial_t e^{-E_\gamma(\tilde{\mathbf{Z}}_t)} = \frac{1}{2}\Delta_{\tilde{\mathbf{Z}}_t} e^{-E_\gamma(\tilde{\mathbf{Z}}_t)}. \tag{27}$$

Using the identity $\Delta(e^{-f}) = (-\Delta f + \|\nabla f\|^2)e^{-f}$ and substituting $f = E_\gamma$ into Eq. 27, we derive the evolution equation for the energy surface:

$$-(\partial_t E_\gamma)e^{-E_\gamma} = \frac{1}{2}\big(-\Delta_{\tilde{\mathbf{Z}}_t} E_\gamma + \|\nabla_{\tilde{\mathbf{Z}}_t} E_\gamma\|^2\big) e^{-E_\gamma}. \tag{28}$$

Dividing by $-e^{-E_\gamma}$, we obtain the Viscous Hamilton-Jacobi equation:

$$\partial_t E_\gamma(\tilde{\mathbf{Z}}_t) = \frac{1}{2}\Delta_{\tilde{\mathbf{Z}}_t} E_\gamma(\tilde{\mathbf{Z}}_t) - \frac{1}{2}\|\nabla_{\tilde{\mathbf{Z}}_t} E_\gamma(\tilde{\mathbf{Z}}_t)\|^2. \tag{29}$$

The ED at scale $t$ is given by:

$$\mathrm{ED}_\gamma(t) = \mathbb{E}_{p_\mathrm{d}(\mathbf{Z})}[E_\theta(\mathbf{Z})] - \int p_t(\tilde{\mathbf{Z}}_t) E_\gamma(\tilde{\mathbf{Z}}_t) \mathrm{d}\tilde{\mathbf{Z}}_t. \tag{30}$$

Taking the derivative w.r.t. $t$, where the first term is constant:

$$\frac{\mathrm{d}}{\mathrm{d}t}\mathrm{ED}_\gamma(t) = -\int (\partial_t p_t) E_\gamma \mathrm{d}\tilde{\mathbf{Z}}_t - \int p_t (\partial_t E_\gamma) \mathrm{d}\tilde{\mathbf{Z}}_t. \tag{31}$$

Since $p_t$ is a convolution with the Gaussian kernel, it satisfies $\partial_t p_t = \frac{1}{2}\Delta_{\tilde{\mathbf{Z}}_t} p_t$. Substituting this into the first term of Eq. 31 and applying integration by parts:

$$\int (\partial_t p_t) E_\gamma \mathrm{d}\tilde{\mathbf{Z}}_t = \frac{1}{2}\int (\Delta_{\tilde{\mathbf{Z}}_t} p_t) E_\gamma \mathrm{d}\tilde{\mathbf{Z}}_t = \frac{1}{2}\int p_t (\Delta_{\tilde{\mathbf{Z}}_t} E_\gamma)\mathrm{d}\tilde{\mathbf{Z}}_t. \tag{32}$$

Next, we substitute Eq. 29 into the second term of Eq. 31:

$$\int p_t(\partial_t E_\gamma)\mathrm{d}\tilde{\mathbf{Z}}_t = \int p_t(\frac{1}{2}\Delta_{\tilde{\mathbf{z}}_t} E_\gamma - \frac{1}{2}\|\nabla_{\tilde{\mathbf{z}}_t} E_\gamma\|^2)\mathrm{d}\tilde{\mathbf{Z}}_t. \tag{33}$$

Combining these results back into Eq. 31:

$$\begin{aligned}
\frac{\mathrm{d}}{\mathrm{d}t}\mathrm{ED}_\gamma(t) &= -\frac{1}{2}\mathbb{E}_{p_t}[\Delta_{\tilde{\mathbf{z}}_t} E_\gamma] - \mathbb{E}_{p_t}[\frac{1}{2}\Delta_{\tilde{\mathbf{z}}_t} E_\gamma - \frac{1}{2}\|\nabla_{\tilde{\mathbf{z}}_t} E_\gamma\|^2] \\
&= \mathbb{E}_{p_t}[\frac{1}{2}\|\nabla_{\tilde{\mathbf{z}}_t} E_\gamma(\tilde{\mathbf{Z}}_t)\|^2 - \Delta_{\tilde{\mathbf{z}}_t} E_\gamma(\tilde{\mathbf{Z}}_t)].
\end{aligned} \tag{34}$$

Eq. 34 is precisely the explicit score matching objective. We integrate Eq. 34 from 0 to $T$ to yield Eq. 26. This confirms that minimizing the ED objective is equivalent to minimizing the accumulated score matching loss across noise scales.

**Monte Carlo Estimation of ED.** We provide a detailed derivation of the practical implementation for the ED objective.

Exploiting the symmetry of the Gaussian kernel, where $\gamma(\tilde{\mathbf{Z}}_t|\mathbf{Z}) = \gamma(\mathbf{Z}|\tilde{\mathbf{Z}}_t)$, we can rewrite Eq. 15 as an expectation over the conditional distribution centered at the perturbed sample:

$$E_\gamma(\tilde{\mathbf{Z}}_t) = -\log \mathbb{E}_{\mathbf{Z}\sim\gamma(\cdot|\tilde{\mathbf{Z}}_t)}[\exp(-E_\theta(\mathbf{Z}))]. \tag{35}$$

By reparameterizing $\mathbf{Z} = \tilde{\mathbf{Z}}_t + \sqrt{t}\boldsymbol{\xi}$, where $\boldsymbol{\xi} \sim \mathcal{N}(\mathbf{0}, \mathbf{I})$, we obtain a form suitable for Monte Carlo estimation:

$$E_\gamma(\tilde{\mathbf{Z}}_t) = -\log \mathbb{E}_{\boldsymbol{\xi}}[\exp(-E_\theta(\tilde{\mathbf{Z}}_t + \sqrt{t}\boldsymbol{\xi}))]. \tag{36}$$

A naive Monte Carlo estimator $\hat{E}_\gamma^{\mathrm{naive}}$ approximates the expectation in Eq. 36 using $M$ i.i.d. samples $\{\boldsymbol{\xi}_j\}_{j=1}^M$:

$$\hat{E}_\gamma^{\mathrm{naive}}(\tilde{\mathbf{Z}}_t) = -\log(\frac{1}{M}\sum_{j=1}^M \exp(-E_\theta(\tilde{\mathbf{Z}}_{t,j}))), \tag{37}$$

where $\tilde{\mathbf{Z}}_{t,j} = \tilde{\mathbf{Z}}_t + \sqrt{t}\boldsymbol{\xi}_j$. Since the function $f(x) = -\log(x)$ is strictly convex, Jensen's inequality implies that this estimator is biased:

$$\mathbb{E}[\hat{E}_\gamma^{\mathrm{naive}}(\tilde{\mathbf{Z}}_t)] = \mathbb{E}[-\log(\frac{1}{M}\sum_{j=1}^M \exp(-E_\theta(\tilde{\mathbf{Z}}_{t,j})))] \geq -\log \mathbb{E}[\exp(-E_\theta(\mathbf{Z}))] = E_\gamma(\tilde{\mathbf{Z}}_t). \tag{38}$$

Thus, the naive estimator systematically overestimates the true contrastive potential. This bias is particularly problematic in high-dimensional spaces where the variance of the exponential importance weights can be extremely large, leading to optimization instability.

Beyond statistical bias, Eq. 37 is numerically fragile. In high-dimensional spaces, perturbed samples $\tilde{\mathbf{Z}}_{t,j}$ often fall into high-energy regions where $\exp(-E_\theta(\tilde{\mathbf{Z}}_{t,j}))$ underflows to 0, causing the logarithm's argument to vanish and gradients to explode. More fundamentally, a high-variance estimator of the contrastive potential can effectively diverge for poorly behaved energies.

To mitigate both the high variance and numerical instability, we introduce a $w$-stabilization term. We augment the estimator with a deterministic anchor derived from the data point $\mathbf{Z}$ and perturbed sample $\tilde{\mathbf{Z}}_t$.

Let $u_0 = E_\theta(\mathbf{Z}) - \log w$ and $u_j = E_\theta(\tilde{\mathbf{Z}}_{t_i,j})$ for $j = 1, \ldots, M$. Since $w \cdot \exp(-E_\theta(\mathbf{Z})) = \exp(-u_0)$, the stabilized estimator can be rewritten as

$$\hat{E}_\gamma^w(\tilde{\mathbf{Z}}_{t_i}) = -\log(\frac{1}{M}\sum_{j=0}^M \exp(-u_j)). \tag{39}$$

Applying the standard soft-min bound[1], we obtain the deterministic upper bound:

$$\hat{E}_\gamma^w(\tilde{\mathbf{Z}}_t) \;\leq\; \min\{E_\theta(\tilde{\mathbf{Z}}_{t,1}), \ldots, E_\theta(\tilde{\mathbf{Z}}_{t,M}), \; E_\theta(\mathbf{Z}) - \log w\} + \log M. \tag{40}$$

Consequently, even if all reconstructed samples underflow or exhibit extreme energies, the anchor term prevents the log-argument from vanishing, providing a controlled and bounded estimation.

Substituting this stabilized estimator into the definition of $\text{ED}_\gamma(t) \approx E_\theta(\mathbf{Z}) - \hat{E}_\gamma^w(\tilde{\mathbf{Z}}_t)$, we derive the single-scale loss:

$$\begin{aligned}
\mathcal{L}_{\text{ED}_\gamma(t)}(\theta) &\approx E_\theta(\mathbf{Z}) - \hat{E}_\gamma^w(\tilde{\mathbf{Z}}_t) \\
&\approx E_\theta(\mathbf{Z}) + \log(\frac{w}{M}\exp(-E_\theta(\mathbf{Z})) + \frac{1}{M}\sum_{j=1}^{M}\exp(-E_\theta(\tilde{\mathbf{Z}}_{t,j}))) \\
&\approx \log(\exp(E_\theta(\mathbf{Z})) \cdot (\frac{w}{M}\exp(-E_\theta(\mathbf{Z})) + \frac{1}{M}\sum_{j=1}^{M}\exp(-E_\theta(\tilde{\mathbf{Z}}_{t,j}))) \\
&\approx \log(\frac{w}{M} + \frac{1}{M}\sum_{j=1}^{M}\exp(E_\theta(\mathbf{Z}) - E_\theta(\tilde{\mathbf{Z}}_{t,j}))).
\end{aligned} \tag{41}$$

Finally, averaging over a multi-scale set of noise levels $\{t_i\}_{i=1}^{S}$ yields the practical objective

$$\mathcal{L}_{\text{ED}}(\theta) \approx \frac{1}{S}\sum_{i=1}^{S}\log(\frac{w}{M} + \frac{1}{M}\sum_{j=1}^{M}\exp(E_\theta(\mathbf{Z}) - E_\theta(\tilde{\mathbf{Z}}_{t_i,j}))). \tag{42}$$

## A.2. Datasets

We evaluate our method on eight widely used benchmark datasets covering citation networks, social media graphs, and e-commerce systems. Table 8 reports key statistics and the standard train/validation/test splits for all datasets. We briefly describe each dataset below:

*Table 8.* Dataset statistics of TAG benchmark datasets.

| CATEGORY | DATASET | #NODES | #EDGES | #NODE DEGREE | #TOKENS | #CLASSES | #SPLIT RATIO (%) |
|---|---|---|---|---|---|---|---|
| Citation | Cora | 2,708 | 5,429 | 4.01 | 186.53 | 7 | 60-20-20 |
| | CiteSeer | 3,186 | 4,277 | 2.68 | 213.16 | 6 | 60-20-20 |
| | PubMed | 19,717 | 44,324 | 4.50 | 468.56 | 3 | 60-20-20 |
| | Arxiv | 169,343 | 1,166,243 | 13.77 | 243.19 | 40 | 54-18-28 |
| Social media | Instagram | 11,339 | 144,010 | 25.40 | 59.25 | 2 | 60-20-20 |
| | Reddit | 33,434 | 198,448 | 11.87 | 203.84 | 2 | 60-20-20 |
| E-commerce | Computer | 87,229 | 721,081 | 16.53 | 90.79 | 10 | 60-20-20 |
| | Photo | 48,362 | 500,928 | 20.72 | 144.59 | 12 | 60-20-20 |

**Cora** (Sen et al., 2008) constitutes a standard academic citation graph comprising 2,708 machine learning publications linked by 5,429 citation edges. Nodes utilize textual attributes derived from paper titles and abstracts while the classification task involves categorizing publications into seven distinct research topics.

**CiteSeer** (Giles et al., 1998) represents a citation network of 3,186 scientific publications primarily from computer science connected by 4,277 links. Textual attributes are generated from paper titles and abstracts to support the classification of documents into six topical categories based on their content and structure.

**PubMed** (Yang et al., 2016) is a biomedical citation graph containing 19,717 publications related to diabetes with 44,324 citation links. Textual attributes are constructed from titles and abstracts and the predictive goal is to classify papers into three specific diabetes-related categories.

---

[1]For $u \in \mathbb{R}^M$, define $\text{LSE}(u) = \log\sum_{j=1}^{M}\exp(u_j)$ and $\text{softmin}(u) = -\text{LSE}(-u)$. Then $\min_j u_j - \log M \leq \text{softmin}(u) \leq \min_j u_j$.

**Arxiv** (Hu et al., 2020) corresponds to Open Graph Benchmark which contains 169,343 computer science papers and 1,166,243 directed citation edges. The task involves predicting the primary subject area from 40 categories using textual attributes generated from titles and abstracts under a chronological evaluation split.

**Instagram** (Huang et al., 2024) functions as a social network graph where 11,339 nodes represent users and 144,010 edges capture following relationships. Textual data includes user profile introductions which facilitate the binary classification of accounts as either commercial or normal users.

**Reddit** (Huang et al., 2024) maps user interactions within discussion communities comprising 33,434 nodes and 198,448 reply-based edges. Texts are aggregated from historical post content to predict user popularity based on whether their average scores exceed the community median.

**Computer** (Yan et al., 2023) originates from the Amazon dataset where 87,229 nodes represent hardware products linked by 721,081 co-purchase relations. Product summaries and reviews serve as textual attributes for classifying items into 10 fine-grained categories within the network.

**Photo** (Yan et al., 2023) is an e-commerce graph of 48,362 photography-related items extracted from Amazon where 500,928 edges reflect frequent co-purchase behaviors. Textual features are constructed from user reviews to classify products into 12 categories based on a specific product taxonomy.

## A.3. Baselines

To ensure a comprehensive comparison, we select representative baselines spanning classic GNNs, Pre-trained Language Models (PLMs), and recent GNN-LLM integrated frameworks, as described below:

**MLP** (Rumelhart et al., 1986) is a foundational feed-forward neural network composed of stacked linear transformations with nonlinear activations. For classification, it maps node features to label logits via a final linear layer. In graph benchmarks, it serves as a structure-agnostic baseline that isolates the predictive value of input features from graph topology.

**GCN** (Kipf & Welling, 2017) learns node representations by propagating and aggregating features over local neighborhoods using a first-order approximation of spectral graph convolutions. Each layer updates embeddings via a normalized adjacency matrix that mixes information from a node and its neighbors.

**GAT** (Veličković et al., 2018) extends neighborhood aggregation with a learnable attention mechanism that assigns different importance weights to neighbors. It computes attention coefficients from node pairs and performs a weighted sum of neighbor features, typically with multi-head attention for stability.

**GraphSAGE** (Hamilton et al., 2017) is an inductive framework that learns parameterized aggregation functions to produce embeddings for unseen nodes. Instead of learning node-specific embeddings, it samples local neighborhoods and aggregates features using trainable operators.

**BERT** (Devlin et al., 2019) is a pre-trained transformer encoder that learns deep bidirectional representations by conditioning on both left and right context. It is trained with masked language modeling and next-sentence prediction to capture general-purpose linguistic features from large corpora. For downstream tasks, it is typically fine-tuned to produce task-specific embeddings or predictions.

**RoBERTa** (Liu et al., 2019) strengthens BERT by optimizing the pre-training recipe. It removes next-sentence prediction, adopts dynamic masking, and trains with larger batches and more data. These changes improve transfer performance across natural language understanding tasks.

**DeBERTa** (He et al., 2021) enhances the transformer by introducing a disentangled attention that separates content and positional representations. It computes attention using distinct projections and employs an improved mask decoder that incorporates absolute position signals. These modifications better capture dependencies and improve contextual modeling.

**OFA** (Liu et al., 2024) is a unified framework for diverse graph classification tasks across domains using a single model interface. It represents graphs as text-attributed graphs and standardizes task specification through a Nodes-of-Interest mechanism. By adding prompting substructures, it supports in-context adaptation and cross-task generalization with limited task-specific fine-tuning.

**GIANT** (Chien et al., 2022) is a self-supervised approach for structure-aware node feature extraction that bridges the gap between graph topology and text. It formulates neighborhood prediction as extreme multi-label classification to fine-tune

language models with structural supervision. The resulting embeddings improve downstream GNN performance by injecting topology-informed semantics.

**TAPE** (He et al., 2024) improves TAG learning by using LLMs to generate natural language explanations for predictions and distilling them into compact node features. A smaller language model converts these explanations into vector representations. A GNN then consumes the enriched features to combine semantic reasoning with structural message passing.

**GLEM** (Zhao et al., 2023) addresses scalability in large TAGs via a variational Expectation-Maximization co-training framework. It alternates between updating an LLM with pseudo-labels and training a GNN with augmented features. This iterative procedure enables mutual reinforcement of textual semantics and graph structure without fully joint optimization.

**SimTeG** (Duan et al., 2023) adopts a decoupled pipeline for efficient textual graph learning. It first applies parameter-efficient fine-tuning to adapt an LLM to the downstream task, then freezes it to generate node embeddings. A lightweight GNN is trained on these embeddings, achieving strong performance with low computational overhead.

**ENGINE** (Zhu et al., 2024) reduces the cost of adapting LLMs to TAGs using a G-Ladder design with lightweight trainable adapters. It injects structural signals into intermediate layers while keeping the LLM backbone frozen to save memory. The framework also supports caching and dynamic early-exit to accelerate training and inference with limited accuracy loss.

**LLaGA** (Chen et al., 2024) adapts LLMs to TAGs by linearizing local neighborhoods into structure-aware node sequences. A learnable projector maps these sequences into the LLM token embedding space, allowing the LLM to process graph structure as token inputs. This enables unified instruction tuning for tasks such as node classification and link prediction without modifying the LLM backbone.

**GraphGPT** (Tang et al., 2024) aligns graph structural knowledge with LLM reasoning via graph-oriented instruction tuning. It introduces a textual grounding component to connect graph tokens with textual semantics and fine-tunes the model to follow graph-specific instructions. This improves generalization to unseen tasks and supports zero-shot inference.

**GraphAdapter** (Huang et al., 2024) injects structural information into a frozen LLM using a lightweight GNN adapter. During language-structure pre-training with an autoregressive objective, the GNN is trained to provide topology-aware context for next-token prediction. This design enables parameter-efficient adaptation for generative graph tasks.

**TEA-GLM** (Wang et al., 2024) enables zero-shot learning by aligning GNN embeddings with the token space of instruction-tuned LLMs. It trains a linear projector that converts graph embeddings into a sequence of soft graph tokens, which are inserted into natural language instructions. This allows the LLM to perform inference on unseen graphs without specific fine-tuning.

### A.4. Prompt Design

In this section, we detail the prompt templates employed by ERAlign$_{\text{LLM}}$. To effectively fuse structural and semantic information, we utilize a soft prompt injection strategy. The aligned GNN representations are mapped to a sequence of $B$ continuous embeddings, denoted as soft tokens $\{\mathbf{H}^{\text{Tok}}\}_B$. These tokens are inserted into the LLM's input embedding space via the placeholder `<graphtokens>`. We adopt a prefix injection strategy, placing `<graphtokens>` immediately following the task header to optimize information flow. We set the sequence length to $B = 10$ to balance representation capacity with computational efficiency.

**Node Classification Prompts.** For node classification, ERAlign is tasked with predicting the correct category from a candidate set. We evaluate the conditional log-probability of each class given the instruction and select the most likely label.

> **Context:** Academic Papers from Citation Networks (Cora, CiteSeer, PubMed, and Arxiv)
> **Input Template:**
> Given the representation of a paper: `<graphtokens>`, with the following information:
> Title: {`title`}
> Abstract: {`abstract`}
> Question: Which subject category does this paper belong to? Please directly give the most likely answer from the following options: {`candidate_labels`}.
> Answer:

**Context:** User Profiles from Instagram
**Input Template:**
Given the representation of a user: `<graphtokens>`, with the following information:
Profile Content: {`profile_intro`}
Question: Is this account a commercial user or a normal user? Choose one from: {`commercial, normal`}.
Answer:

**Context:** Historical Posts from Reddit
**Input Template:**
Given the representation of a user: `<graphtokens>`, with the following information:
Posts Content: {`post_history`}
Question: Is the target popular (average score above the community median) or not popular? Choose one from: {`popular, not_popular`}.
Answer:

**Context:** Product Items from E-commerce Platforms (Computer and Photo)
**Input Template:**
Given the representation of a product: `<graphtokens>`, with the following information:
Product title: {`title`}
Description: {`summary`}
Question: Which specific category does this product belong to? Please directly give the most likely answer from the following options: {`candidate_labels`}.
Answer:

**Link Prediction Prompts.** For link prediction, we formulate the task as a binary query problem. We input the soft tokens and textual description of two nodes and calculate the score difference between affirmative (e.g., "Yes") and negative (e.g., "No") verbalizers.

**Context:** Authorship/Citation Relationship from Citation Networks (Cora, CiteSeer, and Arxiv)
**Input Template:**
Given the representation of two papers:
Paper 1: `<graphtokens_u>` Title: {`title_u`} Abstract: {`abstract_u`}
Paper 2: `<graphtokens_v>` Title: {`title_v`} Abstract: {`abstract_v`}
Question: Do these two papers have a citation relationship? Please answer "Yes" or "No".
Answer:

**Context:** Co-purchase/Co-view interaction from E-commerce Platforms (Computer and Photo)
**Input Template:**
Given the representation of two products:
Product 1: `<graphtokens_u>` Title: {`title_u`} Description: {`summary_u`}
Product 2: `<graphtokens_v>` Title: {`title_v`} Description: {`summary_v`}
Question: Are these two products frequently purchased together? Please answer "Yes" or "No".
Answer:

## A.5. Additional Results

**Node Classification across Varying Supervision Ratios.** We also evaluate node classification performance across varying levels of supervision by using 10% and 40% labeled training nodes on four datasets, as summarized in Table 9. In the 10% labeled setting, ERAlign achieves the best performance with improvements of 3.60% on Cora, 1.21% on CiteSeer, and

2.90% on Photo compared to the strongest baselines. On the Instagram dataset, ERAlign yields the highest result at 63.60%. With 40% labeled nodes, ERAlign$_{GNN}$ further surpasses the leading competitors by margins of 3.19% on Cora, 0.50% on CiteSeer, and 5.81% on Photo. Overall, ERAlign demonstrates robustness across different supervision ratios and shows particularly strong gains on citation and e-commerce graphs. Furthermore, ERAlign$_{GNN}$ becomes increasingly advantageous as the level of supervision rises.

*Table 9.* Node classification performance under various semi-supervised settings with accuracy±std (%) shown. **Bold** and underline highlight the best and second-best results, respectively.

| METHODS | CORA | CITESEER | INSTAGRAM | PHOTO | CORA | CITESEER | INSTAGRAM | PHOTO |
|---|---|---|---|---|---|---|---|---|
| | 10% labeled training nodes | | | | 40% labeled training nodes | | | |
| MLP | 20.43 ± 0.18 | 16.15 ± 0.83 | 62.68 ± 1.03 | 43.43 ± 0.88 | 50.40 ± 0.31 | 49.56 ± 0.43 | 62.51 ± 0.71 | 54.42 ± 0.41 |
| GCN | 31.32 ± 1.14 | 19.74 ± 0.45 | 62.89 ± 0.07 | 43.89 ± 0.74 | 63.84 ± 1.08 | 53.33 ± 0.19 | 63.93 ± 0.58 | 65.71 ± 0.45 |
| GAT | 27.41 ± 0.29 | 19.50 ± 0.62 | 62.65 ± 0.87 | 43.14 ± 1.06 | 62.21 ± 1.05 | 52.91 ± 0.41 | 64.18 ± 0.63 | 65.84 ± 0.32 |
| GraphSAGE | 25.62 ± 0.96 | 20.64 ± 0.76 | 59.11 ± 1.44 | 44.91 ± 0.92 | 61.54 ± 0.47 | 54.66 ± 0.22 | 64.18 ± 0.85 | 64.67 ± 0.06 |
| BERT | 27.73 ± 0.24 | 20.28 ± 0.35 | 62.12 ± 0.39 | 45.13 ± 0.33 | 57.84 ± 0.47 | 53.67 ± 0.64 | 62.93 ± 0.29 | 60.14 ± 0.05 |
| DeBERTa | 18.59 ± 1.53 | 19.47 ± 0.97 | 62.47 ± 0.36 | 42.90 ± 1.66 | 54.89 ± 1.54 | 51.05 ± 0.34 | 64.06 ± 0.24 | 59.34 ± 0.39 |
| RoBERTa | 15.58 ± 0.47 | 12.80 ± 1.46 | 63.02 ± 0.77 | 41.94 ± 0.55 | 53.92 ± 0.68 | 49.65 ± 3.30 | 63.07 ± 0.81 | 64.92 ± 0.21 |
| TAPE | 36.48 ± 0.44 | 18.09 ± 0.39 | 60.62 ± 0.39 | 54.25 ± 0.57 | 68.02 ± 0.45 | 53.52 ± 0.48 | 63.49 ± 0.27 | 71.40 ± 0.63 |
| GLEM | 39.28 ± 0.47 | 27.43 ± 1.59 | 61.24 ± 1.07 | 51.25 ± 2.12 | 67.48 ± 1.08 | 56.97 ± 1.66 | 64.41 ± 0.20 | 66.04 ± 1.62 |
| SimTeG | 34.67 ± 0.57 | 18.33 ± 0.86 | 59.74 ± 0.56 | 49.63 ± 0.65 | 67.09 ± 0.52 | 54.82 ± 0.87 | 63.11 ± 0.32 | 66.85 ± 0.69 |
| ENGINE | 31.16 ± 0.53 | 25.89 ± 0.62 | 63.09 ± 0.56 | 51.85 ± 0.11 | 64.07 ± 0.90 | 55.73 ± 0.50 | 64.61 ± 0.21 | 70.78 ± 0.31 |
| ERAlign$_{LLM}$ | 41.96 ± 1.07 | 28.56 ± 1.28 | 63.34 ± 0.73 | 56.53 ± 1.47 | 70.45 ± 1.24 | 57.10 ± 0.85 | 65.85 ± 0.67 | 74.49 ± 1.27 |
| ERAlign$_{GNN}$ | **42.88** ± 1.14 | **28.64** ± 0.84 | **63.60** ± 0.49 | **57.15** ± 0.31 | **71.21** ± 0.36 | **57.47** ± 0.44 | **65.90** ± 0.63 | **77.21** ± 0.28 |

**Comparison with Recent LLM-based Baselines.** We further compare ERAlign with three recent baselines, including LLM4NG for few-shot node generation (Yu et al., 2025), Locle for label-free node classification (Zhang et al., 2025), and LinkGPT for LLM-enhanced link prediction (He et al., 2025). Since these methods were originally designed for different target tasks with differing data splits, evaluation protocols, and metrics, we carefully modify them to adapt our experimental settings to ensure a fair comparison.

**Full-supervised node classification.** For a fair comparison, we align the standard data splits and evaluation protocols of LLM4NG and Locle with those of ERAlign. As reported in Table 10, ERAlign consistently achieves the best performance across all four datasets. On Cora, CiteSeer, PubMed, and Arxiv, LLM4NG and Locle perform comparably to GraphGPT and LLaGA. This suggests that while simple LLM pseudo-label enhancement or contrastive learning is beneficial, the absence of an explicit distribution alignment mechanism between the GNN and the LLM limits the integration of complementary multimodal information. By employing layer-wise energy-based alignment, ERAlign maps GNN structural representations and LLM semantic representations into a unified space, yielding consistent improvements in node classification.

*Table 10.* Fully-supervised node classification accuracy±std (%) against recent LLM-based baselines. **Bold** indicates the best result.

| METHOD | CORA | CITESEER | PUBMED | ARXIV |
|---|---|---|---|---|
| LLM4NG | 87.24 ± 0.95 | 76.18 ± 1.16 | 89.83 ± 0.41 | 76.28 ± 1.22 |
| Locle | 86.52 ± 1.01 | 75.87 ± 1.21 | 89.22 ± 0.54 | 75.41 ± 0.24 |
| ERAlign$_{LLM}$ | 89.20 ± 1.12 | 77.82 ± 0.50 | 89.55 ± 1.32 | 76.81 ± 0.78 |
| ERAlign$_{GNN}$ | **90.75** ± 0.76 | **78.54** ± 0.34 | **92.17** ± 0.29 | **78.07** ± 0.15 |

**Semi-supervised node classification.** Both LLM4NG and Locle are evaluated using 20% labeled training nodes, with identical to the semi-supervised protocol in Table 2. As shown in Table 11, ERAlign maintains a substantial lead. Although LLM4NG achieves reasonable performance on Cora and CiteSeer via zero-shot LLM annotations, and Locle benefits from contrastive signals, both still fall short of ERAlign. Moreover, the performance gains of LLM4NG and Locle plateau on the Instagram dataset due to weaker textual attributes. In such label-scarce scenarios, ERAlign employs the EBM as a strong regularizer to enforce representation consistency between the GNN and the LLM, effectively propagating structural-semantic signals even under minimal supervision.

*Table 11.* Semi-supervised node classification accuracy±std (%) against recent LLM-based baselines. **Bold** indicates the best result.

| METHOD | CORA | CITESEER | INSTAGRAM | PHOTO |
|---|---|---|---|---|
| LLM4NG | $48.51_{\pm 0.74}$ | $36.92_{\pm 1.12}$ | $60.53_{\pm 0.84}$ | $58.26_{\pm 0.97}$ |
| Locle | $46.82_{\pm 0.85}$ | $35.41_{\pm 0.92}$ | $60.87_{\pm 0.64}$ | $59.14_{\pm 0.71}$ |
| ERAlign$_{\text{LLM}}$ | $51.97_{\pm 0.83}$ | $37.85_{\pm 1.05}$ | $63.96_{\pm 0.95}$ | $61.82_{\pm 0.91}$ |
| ERAlign$_{\text{GNN}}$ | $\mathbf{52.33}_{\pm 0.71}$ | $\mathbf{38.12}_{\pm 0.63}$ | $\mathbf{64.80}_{\pm 0.22}$ | $\mathbf{63.27}_{\pm 0.35}$ |

**Zero-shot cross-task link prediction.** To adapt LinkGPT to our zero-shot transfer protocol, we train it exclusively on node classification and evaluate it directly on link prediction. This adaptation replaces LinkGPT's original link prediction objective with a redesigned node classification module, freezes the trained GNN encoder, and conducts zero-shot inference via LLM prompting. As shown in Table 12, despite being structurally tailored for link prediction with a pairwise encoder, LinkGPT struggles in this cross-task zero-shot setting. While it performs relatively well on Cora and PubMed due to their highly regular linkage patterns, its structural encoder exhibits limited generalization on the Photo dataset. This highlights that robust transferability hinges on learning universal and task-agnostic representations. By constraining the joint distribution via an energy function, ERAlign ensures that node representations inherently preserve rich structural and semantic features, thereby enabling seamless transfer to unseen tasks without task-specific fine-tuning.

*Table 12.* Zero-shot cross-task link prediction AUC (%) against LinkGPT. **Bold** indicates the best result.

| METHOD | CORA | PUBMED | PHOTO |
|---|---|---|---|
| LinkGPT | 57.82 | 68.51 | 52.37 |
| ERAlign$_{\text{LLM}}$ | **60.45** | **71.17** | **57.13** |

**Evaluation on TAPE-Arxiv-23 Dataset.** The widely used Arxiv dataset has been reported by TAPE (He et al., 2024) to potentially suffer from label leakage in LLMs. To avoid this, we further evaluate ERAlign on the newly proposed TAPE-Arxiv-23 dataset (He et al., 2024), which collects arXiv papers released after the knowledge cutoff dates of common LLM corpora. As shown in Table 13, ERAlign$_{\text{GNN}}$ still achieves the highest accuracy, while ERAlign$_{\text{LLM}}$ remains competitive with other strong baselines. These results confirm the robustness of ERAlign's performance gains.

*Table 13.* Node classification accuracy±std (%) on TAPE-Arxiv-23 dataset. **Bold** indicates the best result.

| METHOD | TAPE-ARXIV-23 |
|---|---|
| TAPE-MLP | $83.85_{\pm 2.46}$ |
| TAPE-GCN | $80.80_{\pm 2.15}$ |
| TAPE-SAGE | $83.88_{\pm 2.64}$ |
| TAPE-RevGAT | $84.23_{\pm 2.56}$ |
| ERAlign$_{\text{LLM}}$ | $83.92_{\pm 3.27}$ |
| ERAlign$_{\text{GNN}}$ | $\mathbf{85.11}_{\pm 1.94}$ |

## A.6. Ablation Studies

**Effect of Different LLM Backbones.** Table 14 compares ERAlign$_{\text{LLM}}$ using fixed GraphSAGE layers across Qwen2-7B, LLaMA2-7B, and LLaMA3-8B backbones. In the node classification, LLaMA3-8B achieves competitive average performance and leads on Instagram and Photo with scores of 68.21% and 87.19%, respectively. Conversely, LLaMA2-7B performs slightly better on Cora and CiteSeer with an accuracy of 89.20% and 77.82%, respectively. Link prediction performance depends on the dataset type. Qwen2-7B excels on citation graphs such as Cora and PubMed, whereas LLaMA3-8B outperforms on e-commerce graphs like Photo and Computer. These findings suggest that ERAlign$_{\text{LLM}}$ exhibits reliable transferability across LLM architectures. It is insensitive to specific choices, with stronger backbones providing consistent yet marginal gains.

**Reverse-order Layer Alignment.** To evaluate the impact of layer ordering, we examine a reverse-order alignment scheme where GNN layers are paired with LLM layers in a reverse sequence. For instance, the medium interval $\{4, 12, 20, 28\}$

*Table 14.* Node classification accuracy and link prediction AUC with different LLM backbones. **Bold** indicates the best result.

| BACKBONES | | NODE CLASSIFICATION | | | | LINK PREDICTION | | | |
| --- | --- | --- | --- | --- | --- | --- | --- | --- | --- |
| | | CORA | CITESEER | INSTAGRAM | PHOTO | CORA | PUBMED | PHOTO | COMPUTER |
| ERAlign$_{\text{LLM}}$ w/ GraphSAGE | Qwen2-7B | 88.68 | 76.91 | 66.44 | 87.03 | 60.82 | **71.99** | 56.62 | 58.77 |
| | LLaMA2-7B | **89.20** | **77.82** | 68.05 | 86.95 | 60.45 | 71.17 | 57.13 | 58.63 |
| | LLaMA3-8B | 88.94 | 77.01 | **68.21** | **87.19** | **60.76** | 71.83 | **57.55** | **58.84** |

becomes $\{28, 20, 12, 4\}$. As reported in Table 15, reverse-order alignment consistently underperforms the original sequential matching across all interval densities. This indicates that ERAlign benefits not only from multi-layer constraints but also from preserving the LLM's natural semantic progression from low-level lexical features to high-level semantic abstractions during graph message passing.

*Table 15.* Node classification accuracy (%) of ERAlign$_{\text{GNN}}$ under reverse-order layer alignment on Cora, PubMed, and Photo. "Avg." averages the three datasets. "$\Delta$ vs. Forward" reports the change relative to the corresponding forward strategy in Table 4.

| STRATEGY | LAYER INDEX | CORA | PUBMED | PHOTO | AVG. | $\Delta$ VS. FORWARD |
| --- | --- | --- | --- | --- | --- | --- |
| Reverse sparse | $\{28, 16, 4\}$ | 88.42 | 87.21 | 87.10 | 87.57 | $-0.85$ |
| Reverse medium | $\{28, 20, 12, 4\}$ | 89.63 | 91.18 | 88.27 | 89.69 | $-1.08$ |
| Reverse dense | $\{32, 28, \dots, 8, 4\}$ | 89.07 | 91.56 | 88.11 | 89.58 | $-0.99$ |

**Necessity of the Injection Projector.** We further verify the necessity of the projector $\tilde{\pi}_{\mathcal{G}}$, which injects the aligned LLM semantics back into the GNN backbone. While the shared latent space of dimension $d_{\mathcal{A}}$ serves only to quantify layer-wise alignment, $\tilde{\pi}_{\mathcal{G}}$ maps the aligned LLM semantics back into the GNN's native hidden space of dimension $d_{\mathcal{G}}$ to facilitate structural message propagation. Removing this projector and injecting $\mathbf{Z}_j^{\mathcal{T}}$ directly into the GNN branch enforces $d_{\mathcal{A}}{=}d_{\mathcal{G}}$ and implicitly assumes complete semantic overlap between the two modalities. As shown in Table 16, this restriction bottlenecks the capacity of the alignment space and degrades performance, confirming that $\tilde{\pi}_{\mathcal{G}}$ is essential.

*Table 16.* Node classification accuracy$\pm$std (%) of ERAlign$_{\text{GNN}}$ with and without the projector $\tilde{\pi}_{\mathcal{G}}$. **Bold** indicates the best result.

| STRATEGY | CORA | CITESEER | PUBMED |
| --- | --- | --- | --- |
| Direct Injection ($d_{\mathcal{A}}{=}d_{\mathcal{G}}$) | 88.62 $\pm 0.85$ | 77.14 $\pm 0.68$ | 90.35 $\pm 0.72$ |
| ERAlign$_{\text{GNN}}$ ($d_{\mathcal{A}}{=}512, d_{\mathcal{G}}{=}256$) | **90.75** $\pm 0.76$ | **78.54** $\pm 0.34$ | **92.17** $\pm 0.29$ |

## A.7. Limitations and Future Work

**Limitations.** Our work currently faces the following three main limitations: (i) ERAlign requires access to intermediate LLM hidden states and is therefore primarily applicable to open-weight LLMs. For API-only closed-source LLMs, the full layer-wise alignment is not directly available. (ii) Although the proposed ED scheme substantially improves training efficiency, aligning high-dimensional LLM embeddings still incurs a higher computational cost and memory overhead than training a traditional GNN alone. (iii) ERAlign relies on the presence of informative textual semantics. Its performance gains may diminish on text-irrelevant graphs.

**Future Work.** Several promising directions remain open. First, for closed-source or API-only LLMs whose intermediate hidden states are inaccessible, a practical extension is to impose the energy-based constraint at the output level or prompt interface alone. As suggested by our ablation study, such output-only alignment still enhances cross-modal consistency even without explicit layer-wise supervision. Second, ERAlign currently pairs GNN and LLM layers at fixed intervals. Developing lightweight, memory-efficient mechanisms for adaptive layer matching could reduce the reliance on frequent access to intermediate LLM representations, thereby improving both flexibility and efficiency.

