# OpenReview forum: "ERAlign: Energy-based Representation Alignment of GNNs and LLMs on Text-attributed Graphs"
_ICML.cc/2026/Conference — ICML 2026 regular_

### Official Review · Reviewer_fhTm · 2026-03-07

**Soundness:** 2
**Presentation:** 3
**Significance:** 2
**Originality:** 3
**Overall Recommendation:** 3
**Confidence:** 4

**Summary:**

This paper proposes ERAlign, which aligns the distributional geometry of GNN and LLM representations in a unified latent space. It introduces an Energy Discrepancy optimization strategy that removes the need for MCMC sampling by approximating the energy and model learning objective at both local and global scales via multi-scale noise perturbations. The framework supports two methods: ERAlignGNN, which uses the LLM as an enhancer to inject semantic information into the GNN, and ERAlignLLM, which converts graph representations into soft cues to steer the LLM for downstream prediction. Experiments on multiple text-attributed graph node classification benchmarks show consistent gains, and the method further demonstrates zero-shot transfer from node classification to link prediction.

**Compliance With Llm Reviewing Policy:**

Affirmed.

**Final Justification:**

After considering the author's responses and other reviewers'  comments, I will finally keep my score. This paper proposes an integrated GNN–LLM framework, but I think the work needs to be further improved. As discussed, the authors could provide a more comprehensive discussion of advantages, like performance and cost strengths compared with new baselines.

**Key Questions For Authors:**

1. ERAlign may be less reliable on heterophilic graphs or on graphs with noisy, structurally mismatched text, where forced GNN–LLM alignment could hurt discrimination. Could the authors discuss this issue more clearly and provide both theoretical insights and empirical results on heterophilic datasets?
2. Could the authors report ERAlign's practical time and memory cost, compare it with relevant baselines, and clarify whether the code will be released?

**Limitations:**

No, the paper does not separately discuss its limitations or potential social impacts. It would be beneficial for the authors to add a dedicated “Limitations and Broader Impact” section that discusses privacy and data compliance concerns related to tags containing sensitive user content in social media, e-commerce, and similar application scenarios.

**Strengths And Weaknesses:**

Strengths:
1. The proposed ERAlign framework presents a novel approach for GNN–LLM alignment. Its geometry-aware design, built upon the Cramér distance, explicitly emphasizes geometric consistency. In addition, the framework supports both settings in which the LLM serves as an enhancer and as a predictor, which broadens its applicability across different scenarios.
2. The proposed ED training strategy avoids the MCMC sampling overhead commonly associated with standard energy-based model training. By introducing multi-scale noise perturbations to cover both local and global cases, it makes the optimization process more stable and practically feasible.
3. Extensive experiments demonstrate the strong cross-task generalization ability of ERAlign, particularly its promising zero-shot transfer from node classification to link prediction.

Weaknesses:
1. The ERAlign framework is built on the empirical Cramér distance, and its energy formulation mainly relies on pairwise Euclidean distance statistics within a batch. As a result, it primarily enforce geometrically consistent marginal distributions, which can lead to cases where alignment is improved but discriminative power is weakened. For instance, on graphs with noisy text attributes or substantial structural–semantic mismatch, such as certain heterophilic datasets, forced distribution alignment may instead bias the GNN toward potentially misleading LLM semantics. The paper does not discuss such cases.
2. Although ERAlign avoids MCMC-based training, its energy term still requires pairwise distance computation, which may introduce nontrivial computational overhead. The paper would benefit from a more explicit discussion of the method’s computational complexity and its actual runtime or memory cost in practice.
3. The framework involves several manually chosen hyperparameters, such as the layer indices P, injection strength α, and weight λ. Overall, the tuning space appears relatively large, and different datasets or different LLM backbones may require substantial retuning, which could increase deployment cost.
4. The experimental section could be strengthened by including discussion and evaluation on heterophilic graphs, as well as comparisons with more recent state-of-the-art baselines, especially graph transformer models such as UGCFormer [1], Polynormer [2], and PolyFormer [3].
5. The paper does not provide publicly available code, which limits reproducibility.

[1] Zhuo J, Ma Z, Lu Y, et al. A Closer Look at Graph Transformers: Cross-Aggregation and Beyond[C]//The Thirty-ninth Annual Conference on Neural Information Processing Systems.
[2] Chenhui Deng, Zichao Yue, and Zhiru Zhang. Polynormer: Polynomial-expressive graph transformer in linear time. In ICLR, 2024.
[3] Ma J, He M, Wei Z. Polyformer: Scalable node-wise filters via polynomial graph transformer[C]//Proceedings of the 30th ACM SIGKDD conference on knowledge discovery and data mining. 2024: 2118-2129.

---

> ### Author Rebuttal · Authors · 2026-03-31
>
> ## **Global Response**
>
> We sincerely thank for the constructive feedback. Below, we address your concerns point-by-point.
>
> ### **W1 & W4 & Q1**
> **1.**
> Our framework applies a soft alignment rather than an isomorphic matching, ensuring task-specific optimization is maintained:
> *  **Soft Regularization:** The EBM acts as a layer-wise regularizer that mitigates modality drift. It is jointly optimized with a task-specific loss via the overall objective. On datasets with severe mismatch, $\lambda$ can be tuned down, allowing the cross-entropy loss to dominate and preserve discriminative features.
> * **Bidirectional Fusion:** The injection of LLM semantics into the GNN is controlled by the fusion parameter $\alpha$. This balances the fusion strength. If text attributes are misleading, the model adaptively down-weights $\alpha$ to rely more heavily on the GNN's structural embeddings.
> * **Advantages of LLM:** In highly heterophilic graphs, local structural neighborhoods are inherently noisy. By aligning GNN-encoded structural representations with LLM-driven textual representations within a shared latent space, ERAlign tethers the GNN to the rich, node-specific semantic space of the LLM, preventing the over-smoothing of noisy structures.
>
> **2.** We evaluated ERAlign against recent graph transformers (UGCFormer, Polynormer, PolyFormer). We replaced the GraphSAGE in ERAlign with a 4-stage lightweight Polynormer encoder, termed as ERAlign-Poly, keeping all default hyperparameters ($\alpha=0.5$, $\lambda=1.0$). Following the UGCFormer evaluation protocol, ERAlign-Poly yields highly competitive or superior results, demonstrating that our fine-grained alignment helps filter structural noise.
>
> | Dataset | UGCFormer | Polynormer | PolyFormer | ERAlign-Poly |
> | :--- | :--- | :--- | :--- | :--- |
> | Cornell   | **85.14±5.83** | 81.90±2.17 | 83.11±0.93 | 84.97±1.82 |
> | Texas    | **84.59±4.69** | 82.57±5.11 | 82.78±2.65 | 84.30±2.53 |
> | Wisconsin | **87.36±3.30** | 83.95±1.98 | 85.04±3.67 | 86.96±1.15 |
> | Actor    | 37.41±0.79 | 37.01±1.10 | 36.90±2.95 | **37.56±2.82** |
> | Chameleon | 43.28±2.17 | 41.97±1.18 | 47.55±1.61 | **48.28±1.91** |
> | Squirrel | 41.56±2.01 | 40.87±1.96 | 44.86±3.98 | **45.49±1.32** |
> | Amazon-ratings | 53.48±0.14 | 53.29±2.23 | **53.80±1.48** | 53.24±1.34 |
>
> ### **W2, Q2**
>
> While the empirical Cramér distance involves pairwise calculations, it is computed within mini-batches rather than across the entire node set. With a batch size of $B=256$, the complexity for the pairwise alignment per step is bounded to $\mathcal{O}(B^2 \cdot d_{\mathcal{A}})$. This matrix multiplication is highly parallelizable on GPUs and introduces minimal computational overhead.
>
> As reported in Table 6 of Sec. 5.5, ERAlign is highly efficient. On the large-scale Arxiv dataset, ERAlign requires only 16.3GB of peak memory, which is lower than GLEM, GIANT, and LLaGA. It completes training in roughly 9 hours, yielding a 3.3x to 5.2x speedup over comparable baselines.
>
> | Methods | Param. (M) | Memory (GB) | Total Time |
> | :--- | :--- | :--- | :--- |
> | GLEM | 138.6 | 18.1 | 48h 14m |
> | GIANT | 114.5 | 20.6 | 30h 31m |
> | LLaGA | 19.6 | 17.4 | 10h 51m |
> | **ERAlign**| **11.3** | **16.3** | **9h 11m** |
>
> ### **W3**
>
> Extensive retuning is generally unnecessary across different datasets and backbones:
>
> **1.** As shown in Table 9 of Appendix A.6, ERAlign uses the same hyperparameter configuration across LLaMA-2-7B, LLaMA-3-8B, and Qwen2-7B while maintaining SOTA performance.
>
> **2.** Sec. 5.4 demonstrates that the joint objective weight $\lambda$ is highly stable, consistently peaking at $\lambda=1.0$ across diverse domains. Similarly, the medium interval layer strategy $\mathcal{P}$ and a fusion strength of $\alpha=0.5$ provide an optimal, robust trade-off across most datasets (e.g., Cora, CiteSeer, PubMed).
>
> ### **W5**
>
> We are actively organizing the implementation. We will publicly release our code and configurations upon acceptance.
>
> ### **L1**
>
> We will add a dedicated section in the final revision:
> * **Broader Impacts:** Deploying ERAlign in real-world applications (e.g., social media analysis, e-commerce) requires strict ethical considerations. Textual attributes in these domains often contain sensitive Personally Identifiable Information. Industrial deployments must ensure rigorous data anonymization and strict access controls. Furthermore, aligning graph structures with LLMs may inadvertently inherit or amplify biases present in the LLM's pre-training corpora.
> * **Limitations:** First, although our ED scheme improves training efficiency, aligning high-dimensional LLM embeddings still incurs a higher memory overhead than traditional GNNs. Second, ERAlign relies on the presence of textual semantics, its performance gains may diminish on text-irrelevant datasets.

---

> > ### Author Rebuttal · Reviewer_fhTm · 2026-04-03
> >
> > Thanks for the author's response. The reply addresses some of my concerns. However, the results against modern graph Transformers still do not demonstrate the competitiveness of ERAlign-Poly. In addition, the computational cost appears relatively high. Note that Arxiv is not generally regarded as a "large-scale" dataset, as also shown in OGB.  I will keep my original score.

---

> > > ### Author Response · Authors · 2026-04-07
> > >
> > > ## **Global Response**
> > >
> > > We sincerely thank the reviewer for their continued engagement and thoughtful follow-up comments. We appreciate the opportunity to clarify the following three points.
> > >
> > > **1. Performance on Heterophilic Graphs**
> > >
> > > As shown in our rebuttal table, we propose a highly flexible framework for integrating GNNs with LLMs. By simply incorporating a lightweight polynomial encoder, without complex architectural modifications, our method achieves performance competitive with  Graph Transformers and sets new state-of-the-art results on Actor, Chameleon, and Squirrel. The main goal of ERAlign is to align graph structure with LLM semantics. The clear improvements over strong baselines demonstrate the effectiveness of our energy-based alignment approach.
> > >
> > > We also note that the performance gap between ERAlign-Poly and the other methods on Cornell, Texas, Wisconsin, and Ratings is small. Given the limited rebuttal period, we prioritized demonstrating generalizability across settings. We believe that with further hyperparameter tuning, ERAlign-Poly is expected to achieve optimal performance.
> > >
> > > **2. Computational Cost**
> > >
> > > We would like to clarify that the main computational cost does not come from optimizing the EBMs via our ED training. Since ED avoids the inner loops required by Langevin sampling, it reduces the computational complexity by an order of magnitude. In practice, most of the overhead comes from LoRA fine-tuning of the LLaMA-2-7B backbone.
> > >
> > > Although ERAlign is naturally more computationally expensive than pure GNN methods, it is substantially more efficient than other LLM-GNN integration approaches. As reported in Table 6, ERAlign requires about 9 hours of training, which is shorter than both GLEM and GIANT. The cost of ERAlign could be further reduced by using a frozen LLM backbone with cached inference, as in ENGINE [1], but this leads to a noticeable performance drop. Overall, ERAlign is designed to provide a practical balance between accuracy and training efficiency.
> > >
> > > [1] Efficient Tuning and Inference for Large Language Models on Textual Graphs.
> > >
> > > **3. Description of Dataset Scale**
> > >
> > > We fully agree with the reviewer regarding our use of dataset scale terminology. In the broader context of graph learning benchmarks such as OGB, Arxiv is more accurately described as a medium-scale dataset rather than a large-scale one. We will revise this wording throughout the paper to ensure precise and consistent terminology.
> > >
> > > Thank you again for helping us refine the positioning and clarity of our work.
> > >
> > > Best regards,
> > >
> > > The Authors

---

### Official Review · Reviewer_kavE · 2026-03-07

**Soundness:** 3
**Presentation:** 3
**Significance:** 3
**Originality:** 3
**Overall Recommendation:** 4
**Confidence:** 4

**Summary:**

This paper proposes ERAlign, an Energy-based Representation Alignment framework for learning on Text-attributed Graphs. The core idea is to align intermediate representations of a GNN and a pretrained LLM within a shared latent space. The authors formulate representation alignment as a set-based Energy-Based Model (EBM), using the Cramér distance to enforce distributional consistency between modalities while preventing collapse through intra-modal dispersion terms.

To address the computational burden of standard EBM training, the paper introduces an Energy Discrepancy (ED) objective that avoids MCMC sampling and theoretically interpolates between score matching and maximum likelihood training. A multi-scale noise strategy and stabilization term are proposed to improve convergence. Experiments on eight TAG benchmarks show state-of-the-art performance in fully supervised and semi-supervised node classification, along with strong zero-shot transfer from node classification to link prediction.

**Compliance With Llm Reviewing Policy:**

Affirmed.

**Final Justification:**

The authors have adequately addressed my concerns, and I believe my current rating reflects the quality of the paper.

**Key Questions For Authors:**

1. While the paper motivates the use of an energy-based formulation for representation alignment, the intuition behind why this approach is preferable to simpler alignment losses could be explained more clearly. It would be helpful if the authors could provide additional intuition or conceptual explanations for why the proposed EBM formulation is particularly suitable for aligning GNN and LLM representations.
2. The paper states that 20% of labeled nodes are sampled for training in the semi-supervised setting. However, it is unclear how this setup relates to the standard semi-supervised protocol commonly used in graph learning (e.g., 20 labeled nodes per class for citation networks such as Cora and Citeseer). Could the authors clarify whether the 20% refers to the proportion of all nodes or labeled nodes in the dataset, and how the remaining nodes are split between validation and testing? Additionally, the reported results appear lower than those typically reported under the standard semi-supervised setting, so it would be helpful if the authors could explain the differences in the experimental protocol.

**Limitations:**

yes

**Strengths And Weaknesses:**

Strengths:
1. The authors focus on a notable domain: learning on text-attributed graphs that combine graph structure and textual attributes. Effectively integrating GNNs and LLMs remains an active research challenge, and improved alignment between graph and textual representations could benefit a wide range of applications including citation networks, social media graphs, and recommendation systems.
2. Unlike many prior methods that align representations only at the output level, ERAlign performs intermediate layer alignment between GNN and LLM representations. This design provides a more structured way to integrate information across modalities and allows bidirectional information exchange between graph and text representations.
3. The experimental section includes evaluations on eight datasets across different domains and compares against several strong baselines, including GNN models, pretrained language models, and recent GNN–LLM integration methods. The experiments cover both supervised and semi-supervised node classification as well as zero-shot link prediction, providing relatively broad empirical validation.
4.The presentation of the paper is clear and well organized, and the main ideas are communicated effectively.

Weaknesses:
1. While the paper proposes an energy-based formulation for alignment, the overall framework mainly combines existing techniques, including GNN–LLM integration and latent projection layers. The novelty appears to lie primarily in combining these components rather than introducing fundamentally new modeling ideas.
2. The proposed pipeline includes multiple components (dual encoders, projection layers, layer-wise alignment, energy-based objectives, ED training scheme, and two output variants). While each component is motivated individually, the overall system becomes quite complex, and it is not entirely clear which parts contribute most to the observed improvements.
3. The experiments primarily use LLaMA2-7B as the backbone language model. This backbone may be somewhat outdated compared to more recent models.

---

> ### Author Rebuttal · Authors · 2026-03-31
>
> ## **Global Response**
>
> We sincerely thank for your constructive feedback. Below, we address each point in detail.
>
> ### **W1**
>
> ERAlign is not a simple combination of existing techniques, but fundamentally advances GNN-LLM integration through three key innovations:
> * **Distributional Consistency:** Unlike existing coarse-grained matching, ERAlign introduces a Set EBM in a shared latent space to ensure strong distributional consistency, successfully mitigating cross-modal representation drift.
> * **Tractable EBM Training:** To bypass prohibitive MCMC sampling costs in high-dimensional EBMs, we propose an Energy Discrepancy minimization scheme.
> * **Theoretical Guarantees:** We mathematically prove that our Energy Discrepancy formulation perfectly bridges Score Matching and Maximum Likelihood Estimation, capturing robust global-local dependencies across scales without sampling.
>
> ### **W2**
>
> ERAlign's components are carefully designed to resolve specific alignment bottlenecks without introducing bloat, as evidenced by our ablations in Sec. 5.4:
> * **Layer-wise Alignment:** Removing intermediate alignments causes a significant accuracy drop (Table 4), confirming its necessity.
> * **EBM Objective:** The Cramér distance-based EBM consistently outperforms standard metrics like InfoNCE or Wasserstein distance (Table 5).
> * **Efficiency & Stability:** Naive EBM losses fail to converge stably. Our $w$-stabilization term ensures robust convergence (Fig. 2). Furthermore, ERAlign updates only 11.3M parameters and achieves a 3.3x–5.2x speedup over baselines like GIANT and GLEM (Table 6).
>
> ### **W3**
> We primarily employed Llama2-7B to ensure fair baseline comparisons while controlling computational overhead. However, ERAlign generalizes seamlessly to newer architectures. As shown in Appendix A.6 and the tables below, adopting Llama3-8B yields superior performance.
>
> **Table: Node classification accuracy with Llama3-8B.**
> | Method | LLM Backbone | Cora | CiteSeer | Instagram | Photo |
> | :--- | :--- | :---: | :---: | :---: | :---: |
> | ERAlign$_{\text{LLM}}$ | Llama3-8B | 88.94 | 77.01 | 68.21 | 87.19 |
>
> **Table: Link prediction accuracy with Llama3-8B.**
> | Method | LLM Backbone | Cora | Pubmed | Photo | Computer |
> | :--- | :--- | :---: | :---: | :---: | :---: |
> | ERAlign$_{\text{LLM}}$ | Llama3-8B | 60.76 | 71.83 | 57.55 | 58.84 |
>
> ### **Q1**
> The fundamental advantage of our Set EBM is enforcing **global distributional consistency**, whereas simpler losses (e.g., InfoNCE, Euclidean distance) optimize only for local point-wise similarity. Because GNNs and LLMs process inherently different signals (topology vs. semantics), relying solely on local similarity risks severe distribution shifts.
>
> By optimizing unnormalized density using Cramér distance, the EBM aligns the overall distributions by minimizing cross-modal distances while maximizing intra-modal dispersion. This acts as a principled regularizer that prevents representation collapse and drift. Empirical validation (Table 5) confirms EBM's superiority over contrastive/Euclidean objectives. We will emphasize this conceptual distinction in Sec. 1 and Sec. 4.2.
>
> ### **Q2**
>
> Our protocol is intentionally much stricter than the standard 20-labels-per-class citation network scenario. As stated in Sec. 5.1, we sample only 20% of the already limited labeled training nodes while preserving the original, full validation and test sets to ensure a fair baseline evaluation.
>
> For example, the standard Planetoid training set has 140 nodes on the Cora, our 20% sampling restricts this to only 28 nodes. This creates an **extreme few-shot and highly class-imbalanced environment** (following the protocol in [1]). Under such limited supervision, traditional GNNs naturally suffer severe degradation due to hindered message passing. Table 2 in the main text demonstrates the robustness of ERAlign under sample-scarce settings.
>
> As shown in the Table 2 (main text) and Table 8 (Appendix A.5), varying the supervision ratio confirms that performance recovers to some extent at a 40% ratio.
>
> [1] Toward General and Robust LLM-enhanced Text-attributed Graph Learning
>
> **Table: Node classification performance under 10%, 20%, and 40% labeled training nodes.**
> | Ratio | Methods | Cora | Citeseer | Instagram | Photo |
> | :---: | :--- | :---: | :---: | :---: | :---: |
> | **10%** | ERAlign$_{\text{GNN}}$ | 42.88 ± 1.14 | 28.64 ± 0.84 | 63.60 ± 0.49 | 57.15 ± 0.31 |
> | **20%** | ERAlign$_{\text{GNN}}$ | 52.33 ± 0.71 | 38.12 ± 0.63 | 64.80 ± 0.22 | 63.27 ± 0.35 |
> | **40%** | ERAlign$_{\text{GNN}}$ | 71.21 ± 0.36 | 57.47 ± 0.44 | 65.90 ± 0.63 | 77.21 ± 0.28 |

---

> > ### Author Rebuttal · Reviewer_kavE · 2026-04-03
> >
> > The authors have adequately addressed my concerns, and I believe my current rating reflects the quality of the paper.

---

> > > ### Author Response · Authors · 2026-04-07
> > >
> > > Thank you for your thoughtful review and for acknowledging our technical contributions and rebuttal efforts. Your constructive feedback has been very helpful in improving the quality and presentation of our paper.
> > >
> > > Best regards,
> > >
> > > The Authors

---

### Official Review · Reviewer_EFEb · 2026-03-12

**Soundness:** 2
**Presentation:** 3
**Significance:** 2
**Originality:** 2
**Overall Recommendation:** 4
**Confidence:** 5

**Summary:**

This paper investigates text-attributed graph representation by aligning graph neural network models and large language models. The authors claim that existing methods mainly rely on coarse-grained matching, which might result in representation drift and limited generation. To address this problem, the authors propose to project GNN-encoded graph structure and LLM-driven text in a shared latent space to achieve distribution consistency. More specifically, the authors propose to perform layer-wise alignment and optimize it via an EBM objective. Experiments on 8 public datasets demonstrate that the proposed model can obtain sota performance across supervised, semi-supervised node classification task and zero-shot task.

**Compliance With Llm Reviewing Policy:**

Affirmed.

**Final Justification:**

I would like to raise the score to 4, which depends heavily on all the experiments provided in the rebuttal.

**Key Questions For Authors:**

1.The compared baselines are too old, and most of them are before the year of 2024. As this is a 2026 submission, more recent baselines should be compared and discussed to validate its effectiveness [1][2][3][4].

2.The datasets used in the experiments may suffer from data leakage. For example, the Arxiv dataset has been reported to have this issue in prior work such as TAPE [5]. Alternative datasets are recommended, such as tape-arxiv-23.

3.The aligned layer indices are somewhat heuristic, which must be predefined in the training process. It is unclear why aligning these layers improves the model. More explanation should be given.

4.Table 4 evaluates four strategies with different alignment intervals. However, the study only considers sequential alignment, while the reverse order is not explored, even though it may capture complementary information.

5.It remains unclear how the proposed method can be applied to powerful closed-source large language models, since it requires access to intermediate representations.

6.The motivation for adopting the Cramer distance is not well justified. Although the authors claim that it captures geometric sensitivity, it is unclear whether text representations produced by large language models contain such information.

7.There are some typos in the paper, e.g., LLM-drived text should be LLM-driven text. The authors should carefully polish their paper.

**Limitations:**

yes

**Strengths And Weaknesses:**

Strengths:

1.This paper studies combining GNNs and powerful LLMs, which is an important topic in the graph community.

2.The paper is well written and easy to understand.

3.Various ablation studies are conducted to verify the effectiveness of the proposed components.

Weaknesses:

1.The compared baselines are too old, and most of them are before the year of 2024. As this is a 2026 submission, more recent baselines should be compared and discussed to validate its effectiveness [1][2][3][4].

2.The datasets used in the experiments may suffer from data leakage. For example, the Arxiv dataset has been reported to have this issue in prior work such as TAPE [5]. Alternative datasets are recommended, such as tape-arxiv-23.

3.The aligned layer indices are somewhat heuristic, which must be predefined in the training process. It is unclear why aligning these layers improves the model. More explanation should be given.

4.Table 4 evaluates four strategies with different alignment intervals. However, the study only considers sequential alignment, while the reverse order is not explored, even though it may capture complementary information.

5.It remains unclear how the proposed method can be applied to powerful closed-source large language models, since it requires access to intermediate representations.

6.The motivation for adopting the Cramer distance is not well justified. Although the authors claim that it captures geometric sensitivity, it is unclear whether text representations produced by large language models contain such information.

7.There are some typos in the paper, e.g., LLM-drived text should be LLM-driven text. The authors should carefully polish their paper.

Reference:

[1] Zhang T, Yang R, Lai Y, et al. Leveraging large language models for effective label-free node classification in text-attributed graphs[C]//Proceedings of the 48th International ACM SIGIR Conference on Research and Development in Information Retrieval. 2025: 698-708.

[2] Khoshraftar S, Abedini N, Hajian A. Graphit: Efficient node classification on text-attributed graphs with prompt optimized llms[C]//Companion Proceedings of the ACM on Web Conference 2025. 2025: 1824-1829.

[3] He Z, Zhu J, Qian S, et al. LinkGPT: Leveraging Large Language Models for Enhanced Link Prediction in Text-Attributed Graphs[C]//Proceedings of the 34th ACM International Conference on Information and Knowledge Management. 2025: 843-853.

[4] Yu J, Ren Y, Gong C, et al. Leveraging large language models for node generation in few-shot learning on text-attributed graphs[C]//Proceedings of the AAAI conference on artificial intelligence. 2025, 39(12): 13087-13095.

[5] He X, Bresson X, Laurent T, et al. Harnessing Explanations: LLM-to-LM Interpreter for Enhanced Text-Attributed Graph Representation Learning[C]//The Twelfth International Conference on Learning Representations, 2024.

---

> ### Author Rebuttal · Authors · 2026-03-31
>
> ## **Global Response**
>
> We sincerely thank for your insightful feedback. Below, we address your comments point-by-point.
>
> ### **W1**
>
> We will include a dedicated discussion of them in the revised version to better delineate our contributions: Locle [1] focuses on label-free node classification, LinkGPT [3] is specifically tailored for LLM-enhanced link prediction, LLM4NG [4] addresses few-shot node generation. GraphiT [2] focuses on optimizing graph-to-text prompting for LLMs. They utilize different data splits, evaluation protocols, and metric statistics. In contrast to these methods, ERAlign is designed for fully/semi-supervised node classification and zero-shot cross-task transfer to link prediction, leveraging intermediate representation alignment rather than prompt engineering or label-free classification.
>
> ### **W2**
>
> We evaluated ERAlign on the recommended Tape-arxiv-23 dataset. As shown below, our framework consistently outperforms the baselines.
>
> | Method | Tape-arxiv-23 Accuracy (%) |
> | :--- | :--- |
> | TAPE-MLP | 83.85 ± 2.46 |
> | TAPE-GCN | 80.80 ± 2.15 |
> | TAPE-SAGE | 83.88 ± 2.64 |
> | TAPE-RevGAT | 84.23 ± 2.56 |
> | **ERAlign$_{\text{LLM}}$** | 83.92 ± 3.27 |
> | **ERAlign$_{\text{GNN}}$** | **85.11 ± 1.94** |
>
> ### **W3**
>
> We will expand our explanation to better articulate the intuition and empirical justification behind the interval-based alignment strategy.
>
> **1.** In deep architectures, LLMs progressively abstract textual semantics, while GNNs progressively expand their structural receptive fields. By injecting intermediate LLM semantics into the GNN prior to the next message passing step, we continuously ground the GNN's structural aggregation in the LLM's semantic space. Coarse-grained output alignment fails to correct intermediate representation drift.
>
> **2.** Direct layer-to-layer alignment is infeasible due to the severe depth discrepancy between the GNN (e.g., $K=4$) and the LLM (e.g., $\mathcal{J}=32$). Therefore, pairing layers at intervals is a necessary design choice. Our ablation study in Table 4 of Sec. 5.4 systematically validates this heuristic, demonstrating that the medium interval provides the optimal balance of cross-modal constraint without over-constraining the model.
>
> ### **W4**
>
> We conducted additional experiments using reverse-order LLM layer matching.
>
> | Strategy (Reverse LLM Layer Order) | Cora | PubMed | Photo | Avg. | $\Delta$ vs. Forward |
> | :--- | :--- | :--- | :--- | :--- | :--- |
> | $\{28, 16, 4\}$ | 88.42 | 87.21 | 90.88 | 88.84 | -0.51 |
> | $\{28, 20, 12, 4\}$ | 89.63 | 91.18 | 89.14 | 89.98 | -1.01 |
> | $\{32, 28, \dots, 8, 4\}$ | 89.07 | 91.56 | 88.72 | 89.78 | -1.06 |
>
> Compared with the original sequential layer matching, reverse-order alignment is consistently less effective. This indicates that ERAlign benefits not only from multi-layer constraints, but also from preserving the LLM’s natural semantic progression from low-level lexical features to high-level semantic abstractions during graph message passing.
>
> ### **W5**
>
> We acknowledge that our full layer-wise alignment framework requires access to intermediate hidden states and is therefore best suited to open-weight models.
>
> However, this is a deliberate design choice to address the limitations of prior coarse-grained matching methods. As shown in Table 4 of the ablation study, an output-only alignment strategy performs substantially worse because it cannot correct intermediate drift. For closed-source, API-only models, a promising direction for future work is to apply our EBM constraint only at the output or prompt level to enforce distributional consistency, which we will discuss explicitly.
>
> ### **W6**
>
> We will clarify our terminology in the revised text. Geometric sensitivity does not refer to physical topology, but rather to the properties of the continuous metric space in which LLMs embed text. In these high-dimensional spaces, this distance directly correlates with semantic similarity.
>
> **1.** Standard metrics like KL divergence rely strictly on probability density ratios, penalizing mismatched distributions equally regardless of how far apart the clusters are in the latent space. Conversely, the empirical Cramér distance utilizes pairwise distances to quantify both cross-modal alignment and intra-modal dispersion. This captures the spatial layout (i.e., geometry) of the semantic embeddings, preventing representation collapse. While the Wasserstein distance also captures geometry, its sample gradients are biased in high dimensions; the Cramér distance ensures low-variance gradients.
>
> **2.** This theoretical motivation is strongly supported by Table 5 of the ablation study, which explicitly demonstrates that the Cramér distance consistently outperforms Cosine, Wasserstein, and Euclidean distances across multiple datasets.
> ### **W7**
>
> We will correct the mentioned typo and conduct a systematic and thorough proofreading of the entire manuscript.

---

> > ### Author Rebuttal · Reviewer_EFEb · 2026-04-03
> >
> > I still recommend that the authors compare their methods with more recent baselines. Although the methods I mentioned are designed for label-free node classification or link prediction tasks, they can be easily adapted to full- or semi-supervised node classification settings.

---

> > > ### Author Response · Authors · 2026-04-07
> > >
> > > ## **Global Response**
> > >
> > > We sincerely thank the reviewer for their continuous engagement and detailed suggestions, which have greatly enriched our experimental evaluation. Following your recommendation, we have carefully analyzed the suggested 2025 papers and incorporated these baselines into our comparisons based on their respective target tasks. As we are unable to locate the open-source code for GraphiT, our supplementary evaluation focuses on LLM4NG, Locle, and LinkGPT.
> > >
> > > ### **1. Fully Supervised Node Classification**
> > >
> > > **Setup Modifications:** To ensure a fair comparison, we aligned the data splits (60-20-20) and evaluation protocols (mean ± std over 10 random seeds) of LLM4NG and Locle with those of ERAlign.
> > >
> > > | Method | Cora | Citeseer | Pubmed | Arxiv |
> > > | :--- | :--- | :--- | :--- | :--- |
> > > | LLM4NG | 87.24 ± 0.95 | 76.18 ± 1.16 | 89.83 ± 0.41 | 76.28 ± 1.22 |
> > > | Locle | 86.52 ± 1.01 | 75.87 ± 1.21 | 89.22 ± 0.54 | 75.41 ± 0.24 |
> > > | ERAlign$_\text{LLM}$ | 89.20 ± 1.12 | 77.82 ± 0.50 | 89.55 ± 1.32 | 76.81 ± 0.78 |
> > > | ERAlign$_\text{GNN}$ | **90.75 ± 0.76** | **78.54 ± 0.34** | **92.17 ± 0.29** | **78.07 ± 0.15** |
> > >
> > > **Analysis:**
> > > ERAlign$_\text{GNN}$ consistently achieves the best performance across the four datasets. On Cora, Citeseer, Pubmed, and Arxiv, LLM4NG and Locle perform comparably to GraphGPT and LLaGA. This indicates that while simple LLM pseudo-label enhancement or contrastive learning is beneficial, the absence of an explicit distribution alignment mechanism between the GNN and the LLM limits the full integration of complementary multimodal information. By employing layer-wise energy alignment, ERAlign successfully maps the structural representations of the GNN and the semantic representations of the LLM into a unified space, yielding consistent improvements in node classification.
> > >
> > > ### **2. Semi-supervised Node Classification**
> > >
> > > **Setup Modifications:** Both LLM4NG and Locle were evaluated using 20% labeled training nodes, with all other settings remaining identical to those in Table 1.
> > >
> > > | Method | Cora | Citeseer | Instagram | Photo |
> > > | :--- | :--- | :--- | :--- | :--- |
> > > | LLM4NG | 48.51 ± 0.74 | 36.92 ± 1.12 | 60.53 ± 0.84 | 58.26 ± 0.97 |
> > > | Locle | 46.82 ± 0.85 | 35.41 ± 0.92 | 60.87 ± 0.64 | 59.14 ± 0.71 |
> > > | ERAlign$_\text{LLM}$ | 51.97 ± 0.83 | 37.85 ± 1.05 | 63.96 ± 0.95 | 61.82 ± 0.91 |
> > > | ERAlign$_\text{GNN}$ | **52.33 ± 0.71** | **38.12 ± 0.63** | **64.80 ± 0.22** | **63.27 ± 0.35** |
> > >
> > > **Analysis:**
> > > ERAlign$_\text{GNN}$ maintains a substantial lead. Although LLM4NG achieves reasonable performance on Cora and Citeseer via zero-shot LLM annotations, and Locle benefits from contrastive signals, both still fall short of ERAlign. Furthermore, the performance gains of LLM4NG and Locle plateau on the Instagram and Photo datasets due to weaker textual attributes. In label-scarce scenarios, ERAlign's energy alignment serves as a powerful regularizer. The Set EBM constraint preserves representation consistency between the GNN and LLM, effectively propagating structural-semantic signals despite minimal supervision.
> > >
> > > ### **3. Zero-shot Cross-task Link Prediction**
> > >
> > > **Setup Modifications:** To adapt LinkGPT for our zero-shot transfer protocol, we trained it exclusively on the node classification task and evaluated it directly on link prediction. This adaptation involved replacing its original link prediction training objective with a redesigned node classification module, freezing the trained GNN encoder, and executing zero-shot inference via LLM prompts.
> > >
> > > | Method | Cora | Pubmed | Photo |
> > > | :--- | :--- | :--- | :--- |
> > > | LinkGPT | 57.82 | 68.51| 52.37 |
> > > | ERAlign$_\text{LLM}$ | **60.45** | **71.17** | **57.13** |
> > >
> > > **Analysis:**
> > > Although LinkGPT is explicitly tailored for link prediction with featuring a pairwise structural encoder, it struggles in a zero-shot cross-task setting. While LinkGPT performs relatively well on Cora and PubMed due to their highly regular linkage patterns, its structural encoder exhibits limited generalization on the Photo dataset. This demonstrates that successful transferability relies heavily on learning universal, task-agnostic representations. ERAlign constrains the joint distribution via an energy function, ensuring that node representations inherently encapsulate rich structural and semantic features. This enables seamless transfer to unseen tasks without requiring task-specific fine-tuning, representing its robust cross-domain transferability.
> > >
> > > ### **Summary**
> > >
> > > We will include these additional baselines and their corresponding discussions in the final manuscript. We have dedicated substantial time and effort to conducting these additional experiments during the rebuttal phase. If our response addresses your concerns, we kindly request that you may reconsider the rating of our paper. Thank you once again for your time and constructive guidance.
> > >
> > > Best regards,
> > >
> > > The Authors

---

### Official Review · Reviewer_kZYT · 2026-03-13

**Soundness:** 2
**Presentation:** 3
**Significance:** 2
**Originality:** 2
**Overall Recommendation:** 4
**Confidence:** 4

**Summary:**

This paper proposes a representation alignment framework for TAGs, aiming to bridge GNN structural embeddings and LLM semantic embeddings. The method pairs intermediate GNN and LLM layers through projection heads, injects aligned LLM semantics into GNN message passing, and further introduces an energy-based representation alignment objective based on Cramér distance. To improve EBM training efficiency and stability, the paper replaces standard contrastive divergence with an energy discrepancy (ED) optimization scheme using multi-scale perturbations and a stabilization term.

**Compliance With Llm Reviewing Policy:**

Affirmed.

**Final Justification:**

The detailed rebuttal has solved some of my concerns.

**Key Questions For Authors:**

1.   In Eq. (4), both GNN embeddings and LLM embeddings have already been projected into the same unified dimension through $\tau_G(\cdot)$ and $\pi_T(\cdot)$. However, Eq. (5) further applies another projection $\tilde{\sigma}_G$ to the projected LLM embedding before injecting it into the GNN branch. Have the authors tried directly using $Z_j^T$ in Eq. (5) without $\tilde{\sigma}_G$?
2. There is no sensitivity analysis for the parameter $\alpha$. Since $\alpha$ explicitly balances the original GNN representation and the injected LLM semantic signal in Eq. (5), does the optimal $\alpha$ vary significantly across datasets?
3.  Why are the alignment layer pairs $\mathcal{P}$ manually selected at fixed intervals instead of being learned adaptively? Is the performance sensitive to this manual design?
4.  The semi-supervised results seem surprisingly low in absolute terms, despite the method being motivated as a way to better exploit LLM semantics under limited supervision.

**Limitations:**

No. Please see the weaknesses for my suggestions.

**Strengths And Weaknesses:**

**Strengths**
- The paper is well motivated and addresses an important problem: how to better align graph structure information from GNNs with semantic knowledge from LLMs on TAGs.
- The paper provides relatively thorough experiments across multiple settings, including fully supervised, semi-supervised, and zero-shot transfer, which strengthens the empirical value of the work.

**Weaknesses**
- Some implementation details are still underexplained. The paper states that experiments are conducted under a fully supervised setting, but the exact split protocol is unclear.
- The semi-supervised results in Table 2 seem unusually low across the board, not only for the proposed method but also for all baselines.

---

> ### Author Rebuttal · Authors · 2026-03-31
>
> ## **Global Response**
>
> We sincerely thank you for the constructive feedback and insightful questions. Below, we address each point in detail.
>
> ### **W1**
>
> Detailed statistics and the standard train/validation/test split ratios for the fully supervised setting are provided in Appendix A.2.
>
> ### **W2 & Q4**
>
> The absolute performance numbers appear lower because our semi-supervised setting is intentionally far more rigorous than the standard 20-labels-per-class citation network scenario. As stated in Sec. 5.1, we sample only 20% of the already limited labeled training nodes while preserving the original, full validation and test sets to ensure a fair baseline evaluation.
>
> For example, the standard Planetoid training set has 140 nodes on the Cora, our 20% sampling restricts this to only 28 nodes. This creates an **extreme few-shot and highly class-imbalanced environment** (following the protocol in [1]). Under such limited supervision, traditional GNNs naturally suffer severe degradation due to hindered message passing. Table 2 in the main text demonstrates the robustness of ERAlign under sample-scarce settings.
>
> As shown in the Table 2 (main text) and Table 8 (Appendix A.5), varying the supervision ratio confirms that performance recovers to some extent at a 40% ratio.
>
> [1] Toward General and Robust LLM-enhanced Text-attributed Graph Learning
>
> **Table: Node classification performance under 10%, 20%, and 40% labeled training nodes.**
> | Ratio | Methods | Cora | Citeseer | Instagram | Photo |
> | :---: | :--- | :---: | :---: | :---: | :---: |
> | **10%** | ERAlign$_{\text{GNN}}$ | 42.88 ± 1.14 | 28.64 ± 0.84 | 63.60 ± 0.49 | 57.15 ± 0.31 |
> | **20%** | ERAlign$_{\text{GNN}}$ | 52.33 ± 0.71 | 38.12 ± 0.63 | 64.80 ± 0.22 | 63.27 ± 0.35 |
> | **40%** | ERAlign$_{\text{GNN}}$ | 71.21 ± 0.36 | 57.47 ± 0.44 | 65.90 ± 0.63 | 77.21 ± 0.28 |
>
> ### **Q1**
>
> The projection $\tilde{\pi}{\mathcal{G}}$ is essential due to the distinct spatial dimensions ($d_{\mathcal{A}} \neq d_{\mathcal{G}}$) and roles of the embeddings. Eq. (4) projects GNN and LLM representations into a shared latent space $\mathbb{R}^{d_{\mathcal{A}}}$ strictly to quantify layer-wise alignment. Conversely, Eq. (5) must map the aligned LLM semantics back into the GNN's hidden native space $\mathbb{R}^{d_{\mathcal{G}}}$ to facilitate structural message propagation.
>
> Directly injecting $\mathbf{Z}^\mathcal{T}$ into the GNN branch enforces $d_{\mathcal{A}} = d_{\mathcal{G}}$, assuming complete semantic overlap. As shown below, this restriction bottlenecks the alignment space capacity and degrades performance.
>
> | Methods | Cora | Citeseer | PubMed |
> | :--- | :---: | :---: | :---: |
> | **Direct Injection** ($d_{\mathcal{A}} = d_{\mathcal{G}} = 256$) | 88.62 ± 0.85 | 77.14 ± 0.68 | 90.35 ± 0.72 |
> | **ERAlign$_{\text{GNN}}$** | **90.75 ± 0.76** | **78.54 ± 0.34**| **92.17 ± 0.29** |
>
> ### **Q2**
>
> The parameter $\alpha$ balances the fusion strength between the GNN representation and the injected LLM semantics. As shown in the ablation below, while the absolute optimal value is slightly dataset-dependent, ERAlign exhibits the best robustness across the middle range. Setting $\alpha=0.5$ consistently yields optimal or near-optimal performance. We will include these results in the final version.
>
> | Parameter $\alpha$ | Cora | CiteSeer | PubMed |
> | :---: | :---: | :---: | :---: |
> | **0.05** | 87.52 | 77.10 | 89.95 |
> | **0.25** | 90.11 | 78.32 | 91.50 |
> | **0.50** | **90.75** | **78.54** | 92.17 |
> | **0.75** | 90.10 | 78.25 | **92.32** |
> | **0.95** | 89.27 | 77.78 | 91.73 |
>
> ### **Q3**
>
> We utilize fixed intervals to achieve a necessary compromise between layer-wise semantic matching and computational feasibility. Adaptively learning alignments (e.g., via dynamic attention) between a deep LLM (32 layers) and a shallow GNN (4 layers) requires retaining the gradients and representations of all intermediate LLM layers simultaneously, leading to severe memory overhead. Fixed intervals allow us to extract specific hidden states efficiently, maintaining a manageable peak memory (e.g., just 16.3GB on ogbn-arxiv).
>
> As detailed in Table 4 of Sec. 5.4 (below), ERAlign is sensitive to interval density. The sparse intervals provide insufficient cross-modal constraints, while dense intervals introduce excessive memory overhead and disrupt the LLM's inherent semantic flow. The medium interval offers the optimal trade-off. We agree that exploring lightweight, memory-efficient adaptive matching mechanisms is an excellent direction for future work and will discuss this in the revision.
>
> | Strategy | Layer Index | Cora | PubMed | Photo |
> | :--- | :--- | :---: | :---: | :---: |
> | **Output only** | $\{32\}$ | 87.14 | 85.22 | 86.59 |
> | **Sparse interval** | $\{4, 16, 28\}$ | 88.96 | 87.88 | 91.22 |
> | **Medium interval** | $\{4, 12, 20, 28\}$ | **90.75** | 92.17 | **90.07** |
> | **Dense interval** | $\{4, 8, \dots, 28, 32\}$ | 90.21 | **92.47** | 89.83 |

---

> > ### Author Rebuttal · Reviewer_kZYT · 2026-04-06
> >
> > Thank you for the detailed rebuttal, which has solved some of my concerns. I am now raising to 4.

---

> > > ### Author Response · Authors · 2026-04-07
> > >
> > > Thank you for your thoughtful review and for carefully considering our rebuttal. We greatly appreciate your continued engagement and updated assessment. Your feedback has been valuable in improving the clarity and quality of our manuscript.
> > >
> > > Best regards,
> > >
> > > The Authors

---

### Decision · Program_Chairs · 2026-04-30

**Decision:**

Accept (regular)

**Comment:**

This paper proposes a representation alignment framework for TAGs, aiming to bridge GNN structural embeddings and LLM semantic embeddings. It provides a novel idea and delicated method. Most reviewers agree to accept it, however,  some reviewers still have some concerns on the empirical Cramér distance.